# Continuous Control With Ensemble Deep Deterministic Policy Gradients

## Abstract

The growth of deep reinforcement learning (RL) has brought multiple exciting tools and methods to the field. This rapid expansion makes it important to understand the interplay between individual elements of the RL toolbox. We approach this task from an empirical perspective by conducting a study in the continuous control setting. We present multiple insights of fundamental nature, including: a commonly used additive action noise is not required for effective exploration and can even hinder training; the performance of policies trained using existing methods varies significantly across training runs, epochs of training, and evaluation runs; the critics' initialization plays the major role in ensemble-based actor-critic exploration, while the training is mostly invariant to the actors' initialization; a strategy based on posterior sampling explores better than the approximated UCB combined with the weighted Bellman backup; the weighted Bellman backup alone cannot replace the clipped double Q-Learning. As a conclusion, we show how existing tools can be brought together in a novel way, giving rise to the Ensemble Deep Deterministic Policy Gradients (ED2) method, to yield state-of-the-art results on continuous control tasks from OpenAI Gym MuJoCo. From the practical side, ED2 is conceptually straightforward, easy to code, and does not require knowledge outside of the existing RL toolbox.

## 1 Introduction

Recently, deep reinforcement learning (RL) has achieved multiple breakthroughs in a range of challenging domains (e.g. Silver et al. (2016); Berner et al. (2019); Andrychowicz et al. (2020b); Vinyals et al. (2019)). A part of this success is related to an ever-growing toolbox of tricks and methods that were observed to boost the RL algorithms' performance (e.g. Hessel et al. (2018); Haarnoja et al. (2018b); Fujimoto et al. (2018); Wang et al. (2020); Osband et al. (2019)). This state of affairs benefits the field but also brings challenges related to often unclear interactions between the individual improvements and the credit assignment related to the overall performance of the algorithm Andrychowicz et al. (2020a); Ilyas et al. (2020).

In this paper, we present a comprehensive empirical study of multiple tools from the RL toolbox applied to the continuous control in the OpenAI Gym MuJoCo setting. These are presented in Section 4 and Appendix B. Our insights include:

- The normally distributed action noise, commonly used for exploration, hinders training.
- The current state-of-the-art methods are unstable under several stability criteria.
- The critics' initialization plays a major role in ensemble-based actor-critic exploration, while the training is mostly invariant to the actors' initialization.
- The approximated posterior sampling exploration (Osband et al., 2013) outperforms approximated UCB exploration combined with weighted Bellman backup (Lee et al., 2020).
- The weighted Bellman backup (Lee et al., 2020) can not replace the clipped double Q-Learning (Fujimoto et al., 2018).

To address some of the issues listed above, we introduce the Ensemble Deep Deterministic Policy Gradient (ED2) algorithm[1], see Section 3. ED2 brings together existing RL tools in a novel way: it is

---

[1]Our code is based on SpinningUp (Achiam, 2018). We open-source it at: `https://github.com/ed2-paper/ED2`.

an off-policy algorithm for continuous control, which constructs an ensemble of streamlined versions of TD3 agents and achieves the state-of-the-art performance in OpenAI Gym MuJoCo, substantially improving the results on the two hardest tasks – Ant and Humanoid. Consequently, ED2 does not require knowledge outside of the existing RL toolbox, is conceptually straightforward, and easy to code.

## 2 BACKGROUND

We model the environment as a Markov Decision Process (MDP). It is defined by the tuple $(\mathcal{S}, \mathcal{A}, R, P, \gamma, p_0)$, where $\mathcal{S}$ is a continuous multi-dimensional state space, $\mathcal{A}$ denotes a continuous multi-dimensional action space, $P$ is a transition kernel, $\gamma \in [0, 1)$ stands for a discount factor, $p_0$ refers to an initial state distribution, and $R$ is a reward function. The agent learns a policy from sequences of transitions $\tau = [(s_t, a_t, r_t, s_{t+1}, d)]_{t=0}^{T}$, called episodes or trajectories, where $a_t \sim \pi(\cdot|s_t)$, $s_{t+1} \sim P(\cdot|s_t, a_t)$, $r_t = R(s_t, a_t, s_{t+1})$, $d$ is a terminal signal, and $T$ is the terminal time-step. A stochastic policy $\pi(a|s)$ maps each state to a distribution over actions. A deterministic policy $\mu : \mathcal{S} \to \mathcal{A}$ assigns each state an action.

All algorithms that we consider in this paper use a different policy for collecting data (exploration) and a different policy for evaluation (exploitation). In order to keep track of the progress, the evaluation runs are performed every ten thousand environment interactions. Because of the environments' stochasticity, we run the evaluation policy multiple times. Let $\{R_i\}_{i=1}^{N}$ be a set of (undiscounted) returns from $N$ evaluation episodes $\{\tau_i\}_{i=1}^{N}$, i.e. $R_i = \sum_{r_t \in \tau_i} r_t$. We evaluate the policy using the average test return $\bar{R} = \frac{1}{N} \sum_{i=1}^{N} R_i$ and the standard deviation of the test returns $\sigma = \sqrt{\frac{1}{N-1} \sum_{i=1}^{N} (R_i - \bar{R})^2}$.

We run experiments on four continuous control tasks and their variants, introduced in the appropriate sections, from the OpenAI Gym MuJoCo suite (Brockman et al., 2016) presented in Figure 1. The agent observes vectors that describe the kinematic properties of the robot and its actions specify torques to be applied on the robot joints. See Appendix D for the details on the experimental setup.

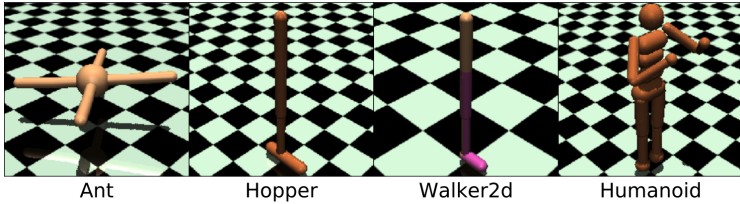

Figure 1: The OpenAI Gym MuJoCo tasks we benchmark out method on.

## 3 ENSEMBLE DEEP DETERMINISTIC POLICY GRADIENTS

For completeness of exposition, we present ED2 before the experimental section. The ED2 architecture is based on an ensemble of Streamlined Off-Policy (SOP) agents (Wang et al., 2020), meaning that our agent is an ensemble of TD3-like agents (Fujimoto et al., 2018) with the action normalization and the ERE replay buffer. The pseudo-code listing can be found in Algorithm 1, while the implementation details, including a more verbose version of pseudo-code (Algorithm 3), can be found in Appendix E. In the data collection phase (Lines 1-9), ED2 selects one actor from the ensemble uniformly at random (Lines 1 and 9) and run its deterministic policy for the course of one episode (Line 4). In the evaluation phase (not shown in Algorithm 1), the evaluation policy averages all the actors' output actions. We train the ensemble every 50 environment steps with 50 stochastic gradient descent updates (Lines 10-13). ED2 concurrently learns $K \cdot 2$ Q-functions, $Q_{\phi_{k,1}}$ and $Q_{\phi_{k,2}}$ where $k \in K$, by mean square Bellman error minimization, in almost the same way that SOP learns its two Q-functions. The only difference is that we have $K$ critic pairs that are initialized with different random weights and then trained independently with the same batches of data. Because of the different initial weights, each Q-function has a different bias in its Q-values. The $K$ actors, $\pi_{\theta_k}$, train maximizing their corresponding first critic, $Q_{\phi_{k,1}}$, just like SOP.

---

**Algorithm 1** ED2 - Ensemble Deep Deterministic Policy Gradients

---

    **Input:** init. params for policy $\theta_k$ and Q-functions $\phi_{k,1}$, $\phi_{k,2}$, $k \in [1...K]$; replay buffer $\mathcal{D}$;
1: Sample the current policy index $c \sim \mathcal{U}([1...K])$.
2: Reset the environment and observe the state $s$.
3: **repeat**
4:     Execute action $a = \mu_{\theta_c}(s)$                     ▷ $\mu$ uses the action normalization
5:     Observe and store $(s, a, r, s', d)$ in the replay buffer $\mathcal{D}$.
6:     Set $s \leftarrow s'$
7:     **if** episode is finished **then**
8:         Reset the environment and observe initial state $s$.
9:         Sample the current policy index $c \sim \mathcal{U}([1...K])$.
10:    **if** time to update **then**
11:       **for** as many as steps done in the environment **do**
12:         Sample a batch of transitions $B = \{(s, a, r, s', d)\} \subset \mathcal{D}$     ▷ uses ERE
13:         Update the parameters $\theta_k$, $\phi_{k,1}$ and $\phi_{k,2}$ by one gradient step.
14: **until** convergence

---

Utilizing the ensembles requires several design choices, which we summarize below. The ablation study of ED2 elements is provided in Appendix C.

**Ensemble**

*Used:* We train the ensemble of 5 actors and 5 critics; each actor learns from its own critic and the whole ensemble is trained on the same data.

*Not used:* We considered different actor-critic configurations, initialization schemes and relations, as well as the use of random prior networks (Osband et al., 2018), data bootstrap (Osband et al., 2016), and different ensemble sizes. We also change the SOP network sizes and training intensity instead of using the ensemble. Besides the prior networks in some special cases, these turn out to be inferior as shown in Section 4 and Appendix B.1.

**Exploration**

*Used:* We pick one actor uniformly at random to collect the data for the course of one episode. The actor is deterministic (no additive action noise is applied). These two choices ensure coherent and temporally-extended exploration similarly to Osband et al. (2016).

*Not used:* We tested several approaches to exploration: using the ensemble of actors, UCB (Lee et al., 2020), and adding the action noise in different proportions. These experiments are presented in Appendix B.2.

**Exploitation**

*Used:* The evaluation policy averages all the actors' output actions to provide stable performance.

*Not used:* We tried picking an action with the biggest value estimate (average of the critics' Q-functions) in evaluation (Huang et al., 2017).

Interestingly, both policies had similar results, see Appendix B.3.

**Action normalization**

*Used:* We use the action normalization introduced by Wang et al. (2020).

*Not used:* We experimented with the observations and rewards normalization, which turned out to be unnecessary. The experiments are presented in Appendix B.4.

**Q-function updates**

*Used:* We do 50 SGD updates (ADAM optimizer (Kingma and Ba, 2015), MSE loss) to the actors and the critics every 50 environment interactions, use Clipped Double Q-Learning (Fujimoto et al., 2018).

*Not used:* We also examined doing the updates at the end of each episode (with the proportional number of updates), using the Hubert loss, and doing weighted Bellman backups (Lee et al., 2020). However, we found them to bring no improvement to our method, as presented in Appendix B.5.

# 4 EXPERIMENTS

In this section, we present our comprehensive study and the resulting insights. The rest of the experiments verifying that our design choices perform better than alternatives are in Appendix B. Unless stated otherwise, a solid line in the figures represents an average, while a shaded region shows a 95% bootstrap confidence interval. We used 30 seeds for ED2 and the baselines and 7 seeds for the ED2 variants.

## 4.1 THE NORMALLY DISTRIBUTED ACTION NOISE, COMMONLY USED FOR EXPLORATION, HINDERS TRAINING

In this experiment, we deprive SOP of its exploration mechanism, namely additive normal action noise, and call this variant deterministic SOP (det. SOP). It causes relatively minor deterioration in the Humanoid performance, has no significant influence on the Hopper or Walker performance, and substantially improves the Ant performance, see Figure 2. This result shows that no additional exploration mechanism, often in a form of an exploration noise (Lillicrap et al., 2016; Fujimoto et al., 2018; Wang et al., 2020), is required for the diverse data collection and it can even hinder training.

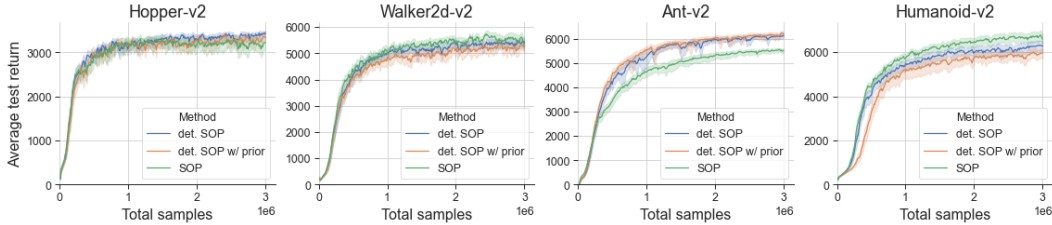

Figure 2: The average test returns across the training of SOP and SOP without the exploration noise in two variants: without and with the prior network. All metrics were computed over 30 seeds.

ED2 leverages this insight and constructs an ensemble of deterministic SOP agents presented in Section 3. Figure 3 shows that ED2 magnifies the beneficial effect coming from the deterministic exploration.

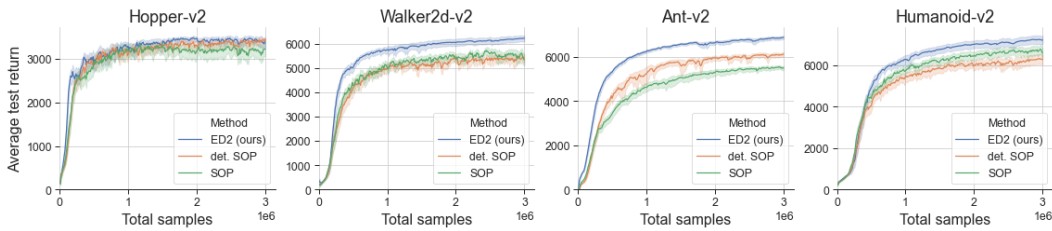

Figure 3: The average test returns across the training of ED2, SOP and SOP without the exploration noise and without the prior network. All metrics computed over 30 seeds.

ED2 achieves state-of-the-art performance on the OpenAI Gym MuJoCo suite. Figure 4 shows the results of ED2 contrasted with three strong baselines: SUNRISE (Lee et al., 2020), SOP (Wang et al., 2020), and SAC(Haarnoja et al., 2018b).

For completeness, we plot the Humanoid velocities in Figure 5 which shows that our method accelerates to a much higher velocity than the baselines.

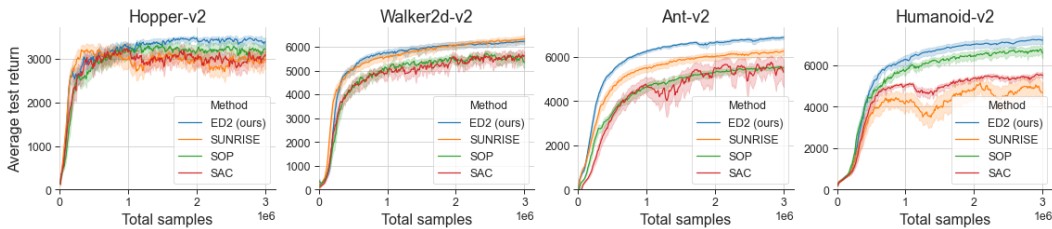

Figure 4: The average test returns across the training of ED2 and the three baselines. The solid lines represent averages, whereas the shaded regions show 95% bootstrap confidence intervals, each being computed over 30 seeds.

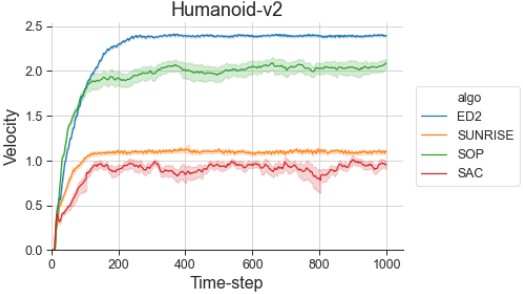

Figure 5: The Humanoid velocity of ED2 and the three baselines. The solid line represents the average velocity and the shaded region is 95% bootstrap confidence interval over 30 runs of the trained agents with the median performance across 30 seeds.

## 4.2 THE CURRENT STATE-OF-THE-ART METHODS ARE UNSTABLE UNDER SEVERAL STABILITY CRITERIA

We consider three notions of stability: inference stability, asymptotic performance stability, and training stability. ED2 outperforms baselines in each of these notions, as discussed below. Similar metrics were also studied in Chan et al. (2020).

**Inference stability**    We say that an agent is *inference stable* if, when run multiple times, it achieves similar test performance every time. We measure inference stability using the standard deviation of test returns explained in Section 2. We found that that the existing methods train policies that are surprisingly sensitive to the randomness in the environment initial conditions[2]. Figure 4 and Figure 6 show that ED2 successfully mitigates this problem. By the end of the training, ED2 produces results within 1% of the average performance on Humanoid, while the performance of SUNRISE, SOP, and SAC may vary as much as 11%.

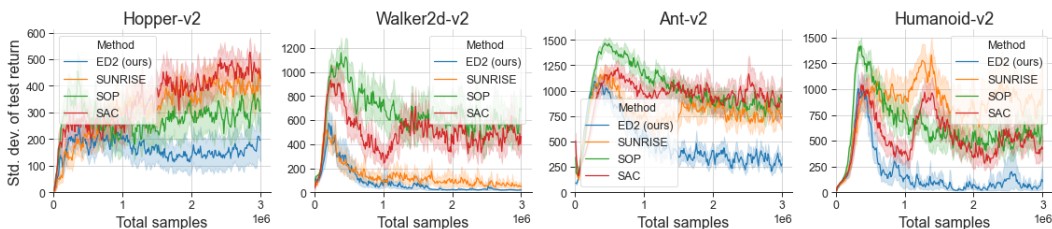

Figure 6: The standard deviation of test returns across training (lower is better), see Section 2.

---

[2]The MuJoCo suite is overall deterministic, nevertheless, little stochasticity is injected at the beginning of each trajectory, see Appendix D for details.

**Asymptotic performance stability**   We say that an agent achieves *asymptotic performance stability* if it achieves similar test performance across multiple training runs starting from different initial networks weights. Figure 7 shows that ED2 has a significantly smaller variance than the other methods while maintaining high performance.

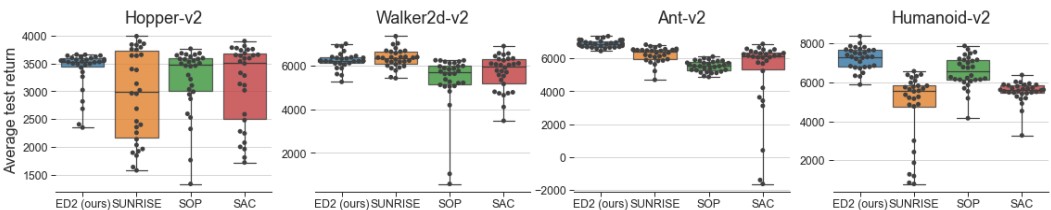

Figure 7: The dots are the average test returns after training ($3M$ samples) of each seed. The distance between each box's top and bottom edges is the interquartile range (IQR). The whiskers spread across all values.

**Training stability**   We will consider training stable if performance does not severely deteriorate from one evaluation to the next. We define the root mean squared deterioration metric (RMSD) as follows:

$$\text{RMSD} = \sqrt{\frac{1}{M}\sum_{i=1}^{M}\Big(\max(\bar{R}_{i-20} - \bar{R}_i, 0)\Big)^2},$$

where $M$ is the number of the evaluation phases during training and $\bar{R}_i$ is the average test return at the $i$-th evaluation phase (described in Section 2). We compare returns 20 evaluation phases apart to ensure that the deterioration in performance doesn't stem from the evaluation variance. ED2 has the lowest RMSD across all tasks, see Figure 8.

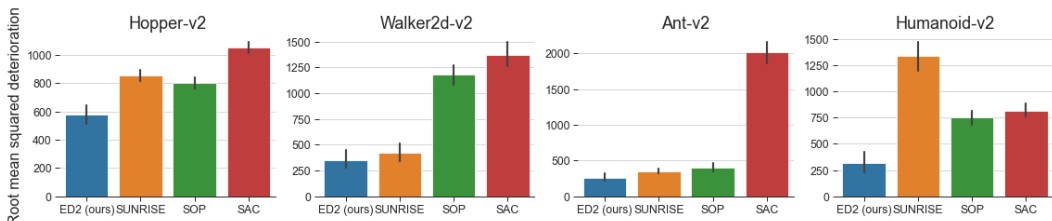

Figure 8: RMSD, the average and the 95% bootstrap confidence interval over 30 seeds.

### 4.3   THE CRITICS' INITIALIZATION PLAYS A MAJOR ROLE IN ENSEMBLE-BASED ACTOR-CRITIC EXPLORATION, WHILE THE TRAINING IS MOSTLY INVARIANT TO THE ACTORS' INITIALIZATION

In this experiment, actors' weights are initialized with the same random values (contrary to the standard case of different initialization). Moreover, we test a corresponding case with critics' weights initialized with the same random values or simply training only a single critic.

Figure 9 indicates that the choice of actors initialization does not matter in all tasks but Humanoid. Although the average performance on Humanoid seems to be better, it is also less stable. This is quite interesting because the actors are deterministic. Therefore, the exploration must come from the fact that each actor is trained to optimize his own critic.

On the other, Figure 9 shows that the setup with the single critic severely impedes the agent performance. We suspect that using the single critic impairs the agent exploration capabilities as its actors' policies, trained to maximize the same critic's $Q$-function, become very similar.

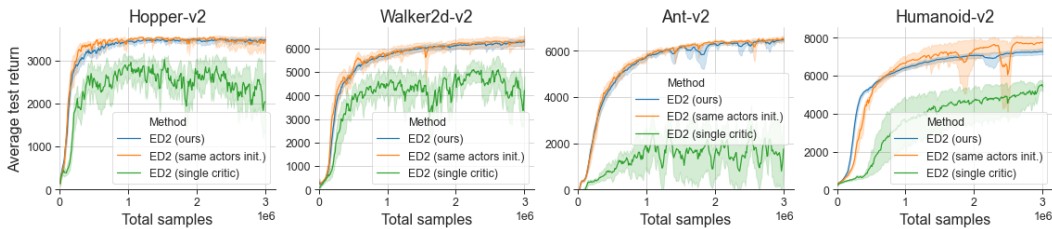

Figure 9: The average test returns across the training of ED2, ED2 with actors initialized to the same random values, and ED2 with the single critic.

### 4.4 THE APPROXIMATED POSTERIOR SAMPLING EXPLORATION OUTPERFORMS APPROXIMATED UCB EXPLORATION COMBINED WITH WEIGHTED BELLMAN BACKUP

ED2 uses posterior sampling based exploration method (Osband et al., 2016). SUNRISE, on the other hand, approximates the Upper Confidence Bound (UCB) exploration technique and does weighted Bellman backups (Lee et al., 2020). For the fair comparison between ED2 and SUNRISE, we substitute the SUNRISE base algorithm SAC for the SOP algorithm used by ED2. We call this variant SUNRISE-SOP.

We test both methods on the standard MuJoCo benchmarks as well as delayed (Zheng et al., 2018a) and sparse (Plappert et al., 2018) rewards variants. Both variations make the environments harder from the exploration standpoint. In the delayed version, the rewards are accumulated and returned to the agent only every 10 time-steps. In the sparse version, the reward for the forward motion is returned to the agent only after it crosses the threshold of one unit on the x-axis. For a better perspective, a fully trained Humanoid is able to move to around five units until the end of the episode. All the other reward components (living reward, control cost, and contact cost) remain unchanged. The results are presented in Table 1.

| Environment | SOP | SUNRISE-SOP | ED2 | Improvement over SOP | Improvement over SUNRISE-SOP |
|---|---|---|---|---|---|
| Hopper-v2 | 3350 | 3083 | 3512 | $+5\%$ | $+14\%$ |
| Walker-v2 | 5458 | 5730 | 6222 | $+14\%$ | $+9\%$ |
| **Ant-v2** | **5507** | **771** | **6862** | $+25\%$ | $+790\%$ |
| **Humanoid-v2** | **6456** | **5738** | **7307** | $+13\%$ | $+27\%$ |
| DelayedHopper-v2 | 2990 | 2708 | 3315 | $+11\%$ | $+22\%$ |
| DelayedWalker-v2 | 4907 | 4505 | 5940 | $+21\%$ | $+32\%$ |
| DelayedAnt-v2 | 4216 | 1486 | 3611 | $-14\%$ | $+43\%$ |
| **DelayedHumanoid-v2** | **770** | **1129** | **3823** | $+397\%$ | $+239\%$ |
| SparseHopper-v2 | 2798 | 2644 | 3397 | $+21\%$ | $+29\%$ |
| SparseWalker-v2 | 4830 | 4664 | 5936 | $+23\%$ | $+27\%$ |
| SparseAnt-v2 | 5573 | **3625** | **5466** | $-2\%$ | $+51\%$ |
| **SparseHumanoid-v2** | **5433** | **5225** | **7438** | $+37\%$ | $+42\%$ |

Table 1: The test returns after training (3M samples) median across 30 seeds for the standard MuJoCo and 7 seeds for the delayed/sparse variants.

ED2 outperforms the non-ensemble method SOP, supporting the argument of coherent and temporally-extended exploration of ED2. Moreover, we observe that performance in MuJoCo environments benefits from the ED2 approximate Bayesian posterior sampling exploration (Osband et al., 2013) in contrast to the approximated UCB in SUNRISE, which follows the OFU principle. The posterior sampling is proved to be theoretically superior to the OFU strategy (Osband and Van Roy, 2017).

The experiment where the ED2's exploration mechanism is replaced for UCB is in Appendix B.2. This variant also achieves worse results than ED2. The additional exploration efficiency experiment in the custom Humanoid environment, where an agent has to find and reach a goal position, is in Appendix A.

### 4.5 THE WEIGHTED BELLMAN BACKUP CAN NOT REPLACE THE CLIPPED DOUBLE Q-LEARNING

We applied the weighted Bellman backups proposed by Lee et al. (2020) to our method. It is suggested that the method mitigates error propagation in Q-learning by re-weighting the Bellman backup based on uncertainty estimates from an ensemble of target Q-functions (i.e. variance of predictions). Interestingly, Figure 10 does not show this positive effect on ED2.

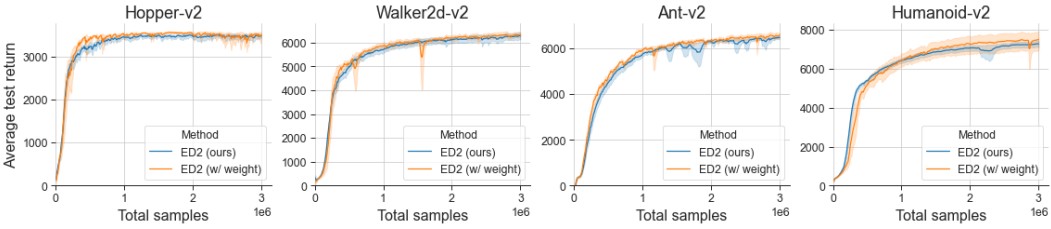

Figure 10: The average test returns across the training of our method and ED2 with the weighted Bellman backup.

Our method uses clipped double $Q$-Learning to mitigate overestimation in $Q$-functions (Fujimoto et al., 2018). We wanted to check if it is required and if it can be exchanged for the weighted Bellman backups used by Lee et al. (2020). Figure 11 shows that clipped double Q-Learning is required and that the weighted Bellman backups can not replace it.

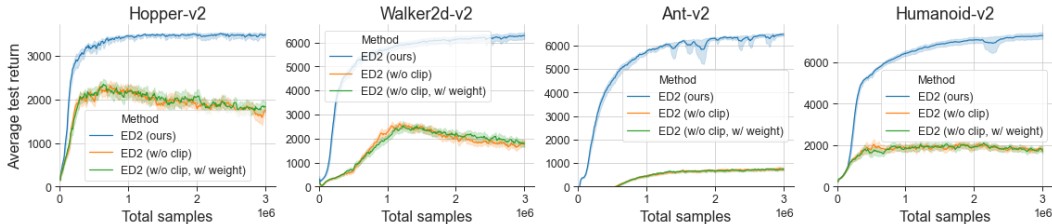

Figure 11: The average test returns across the training of our method and ED2 without clipped double Q-Learning in two variants without and with the weighted Bellman backups.

## 5 RELATED WORK

**Off-policy RL** Recently, multiple deep RL algorithms for continuous control have been proposed, e.g. DDPG (Lillicrap et al., 2016), TD3 (Fujimoto et al., 2018), SAC (Haarnoja et al., 2018b), SOP (Wang et al., 2020), SUNRISE (Lee et al., 2020). They provide a variety of methods for improving training quality, including double-Q bias reduction van Hasselt et al. (2016), target policy smoothing or different update frequencies for actor and critic Fujimoto et al. (2018), entropy regularization Haarnoja et al. (2018b), action normalization Wang et al. (2020), prioritized experience replay Wang et al. (2020), weighted Bellman backups Kumar et al. (2020); Lee et al. (2020), and use of ensembles Osband et al. (2019); Lee et al. (2020); Kurutach et al. (2018); Chua et al. (2018).

**Ensembles** Deep ensembles are a practical approximation of a Bayesian posterior, offering improved accuracy and uncertainty estimation Lakshminarayanan et al. (2017); Fort et al. (2019). They

inspired a variety of methods in deep RL. They are often used for temporally-extended exploration; see the next paragraph. Other than that, ensembles of different TD-learning algorithms were used to calculate better $Q$-learning targets (Chen et al., 2018). Others proposed to combine the actions and value functions of different RL algorithms Wiering and van Hasselt (2008) or the same algorithm with different hyper-parameters Huang et al. (2017). For mixing the ensemble components, complex self-adaptive confidence mechanisms were proposed in Zheng et al. (2018b). Our method is simpler: it uses the same algorithm with the same hyper-parameters without any complex or learnt mixing mechanism. Lee et al. (2020) proposed a unified framework for ensemble learning in deep RL (SUNRISE) which uses bootstrap with random initialization Osband et al. (2016) similarly to our work. We achieve better results than SUNRISE and show in Appendix B that their UCB exploration and weighted Bellman backups do not aid our algorithm performance.

**Exploration**   Various frameworks have been developed to balance exploration and exploitation in RL. The optimism in the face of uncertainty principle Lai and Robbins (1985); Bellemare et al. (2016) assigns an overly optimistic value to each state-action pair, usually in the form of an exploration bonus reward, to promote visiting unseen areas of the environment. The maximum entropy method Haarnoja et al. (2018a) encourages the policy to be stochastic, hence boosting exploration. In the parameter space approach Plappert et al. (2018); Fortunato et al. (2018), noise is added to the network weights, which can lead to temporally-extended exploration and a richer set of behaviours. Posterior sampling Strens (2000); Osband et al. (2016; 2018) methods have similar motivations. They stem from the Bayesian perspective and rely on selecting the maximizing action among sampled and statistically plausible set of action values. The ensemble approach Lowrey et al. (2018); Miłoś et al. (2019); Lee et al. (2020) trains multiple versions of the agent, which yields a diverse set of behaviours and can be viewed as an instance of posterior sampling RL.

## 6   CONCLUSIONS

We conduct a comprehensive empirical analysis of multiple tools from the RL toolbox applied to the continuous control in the OpenAI Gym MuJoCo setting. We believe that the findings can be useful to RL researchers. Additionally, we propose Ensemble Deep Deterministic Policy Gradients (ED2), an ensemble-based off-policy RL algorithm, which achieves state-of-the-art performance and addresses several issues found during the aforementioned study.

## 7   REPRODUCIBILITY STATEMENT

We have made a significant effort to make our results reproducible. We use 30 random seeds, which is above the currently popular choice in the field (up to 5 seeds). Furthermore, we systematically explain our design choices in Section 3 and we provide a detailed pseudo-code of our method in Algorithm 3 in the Appendix B. Additionally, we open-sourced the code for the project[3] together with examples of how to reproduce the main experiments. The implementation details are explained in Appendix E and extensive information about the experimental setup is given in Appendix D.

---

[3]https://github.com/ed2-paper/ED2

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

## A    EXPLORATION EFFICIENCY IN THE CUSTOM HUMANOID ENVIRONMENT

To check the exploration capabilities of our method, we constructed two environments based on Humanoid where the goal is not only to move forward as fast as possible but to find and get to the specific region. The environments are described in Figure 12.

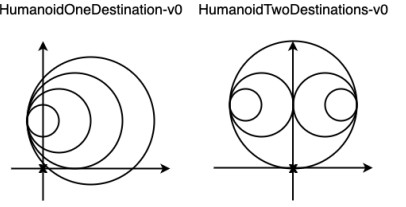

Figure 12: This is a top-down view. Humanoid starts at the origin (marked 'x'). The reward in each time-step is equal to the number of circles for which the agent is inside. Being in the most nested circle (or either of two) solves the task.

Figure 13: The fraction of episodes in which the task is finished for ED2 and two baselines. The average and the 95% bootstrap confidence interval over 20 seeds.

Because the Humanoid initial state is slightly perturbed every run, we compare solved rates over multiple runs, see details in Appendix D. Figure 13 compares the solved rates of our method and the three baselines. Our method outperforms the baselines. For this experiment, our method uses the prior networks (Osband et al., 2018).

## B    DESIGN CHOICES

In this section, we summarize the empirical evaluation of various design choices grouped by topics related to an ensemble of agents (B.1), exploration (B.2), exploitation (B.3), normalization (B.4), and $Q$-function updates (B.5). In the plots, a solid line and a shaded region represent an average and a 95% bootstrap confidence interval over 30 seeds in case of ED2 (ours) and 7 seeds otherwise. All of these experiments test ED2 presented in Section 3 with Algorithm 2 used for evaluation (the ensemble critic variant). We call Algorithm 2 a 'vote policy'.

---

**Algorithm 2** Vote policy

---

1: **Input:** ensemble size $K$; policy $\theta_k$ and $Q$-function $\phi_{k,1}$ parameters where $k \in [1, \ldots, K]$; max action scale $M$;
2: **function** VOTE_POLICY$(s, c)$

$$a_k = M \tanh\left(\mu_{\theta_k}(s)\right) \quad \text{for } k \in [1, \ldots, K] \tag{1}$$

3: **if** use arbitrary critic **then**

$$q_k = Q_{\phi_{c,1}}(s, a_k) \quad \text{for } k \in [1, \ldots, K] \tag{2}$$

4: **else** use ensemble critic

$$q_k = \frac{1}{K} \sum_{i \in [1 \ldots K]} Q_{\phi_{i,1}}(s, a_k) \quad \text{for } k \in [1, \ldots, K] \tag{3}$$

5: **return** $a_k$ for $k = \mathrm{argmax}_k \, q_k$

---

### B.1 ENSEMBLE

**Prior networks**   We tested if our algorithm can benefit from prior networks (Osband et al., 2018). It turned out that the results are very similar on OpenAI Gym MuJoCo tasks, see Figure 14. However, the prior networks are useful on our crafted hard-exploration Humanoid environments, see Figure 15.

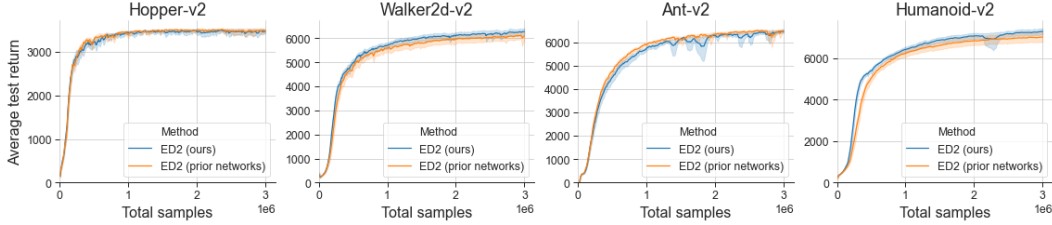

Figure 14: The average test returns across the training of ED2 without (ours) and with prior networks.

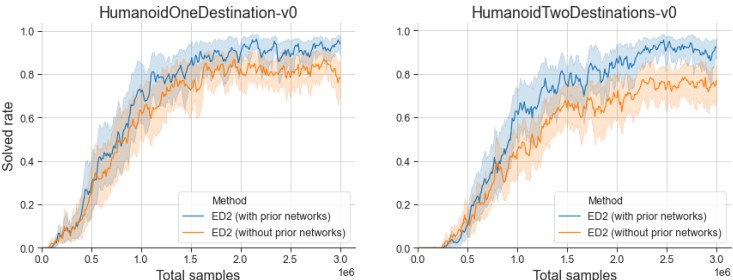

Figure 15: The average test returns across the training of ED2 without (ours) and with prior networks.

**Ensemble size**   Figure 16 shows ED2 with different ensemble sizes. As can be seen, the ensemble of size 5 (which we use in ED2) achieves good results, striking a balance between performance and computational overhead.

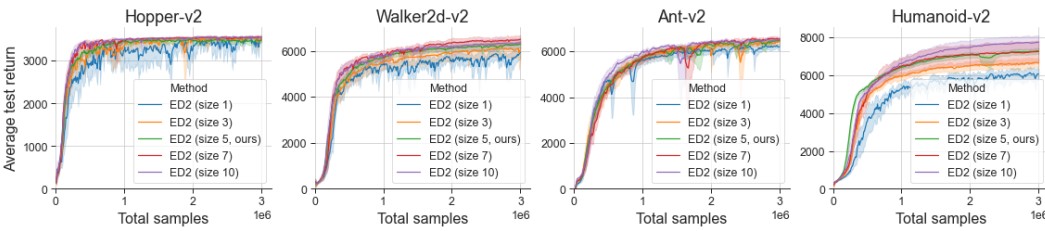

Figure 16: The average test returns across the training of ED2 with a different number of actor-critics.

**Data bootstrap**   Osband et al. (2016) and Lee et al. (2020) remark that training an ensemble of agents using the same training data but with different initialization achieves, in most cases, better performance than applying different training samples to each agent. We confirm this observation in Figure 17. Data bootstrap assigned each transition to each agent in the ensemble with 50% probability.

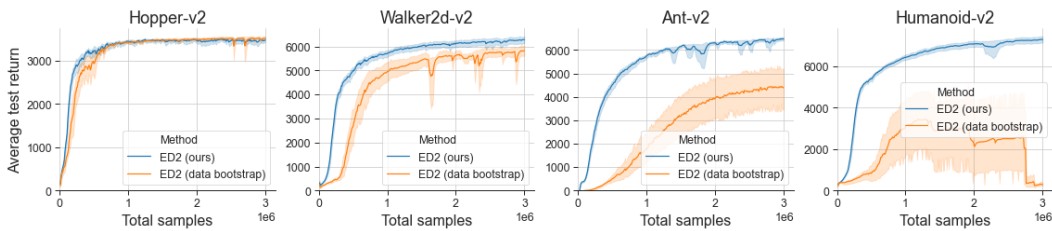

Figure 17: The average test returns across the training of ED2 without (ours) and with data bootstrap.

**SOP bigger networks and training intensity** We checked if simply training SOP with bigger networks or with higher training intensity (a number of updates made for each collected transition) can get it close to the ED2 results. Figure 18 compares ED2 to SOP with different network sizes, while Figure 19 compares ED2 to SOP with one or five updates per environment step. It turns out that bigger networks or higher training intensity does not improve SOP performance.

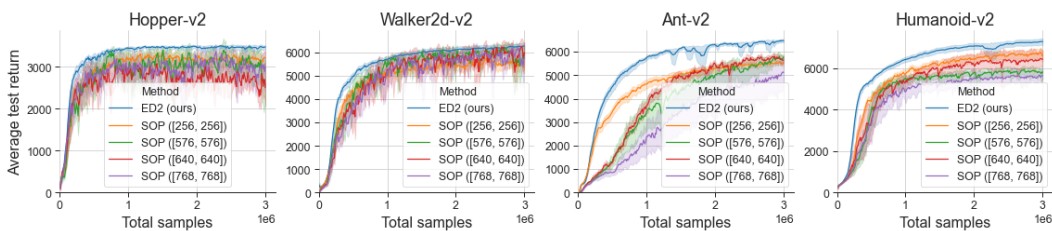

Figure 18: The average test returns across the training of ED2 and SOP with different network sizes.

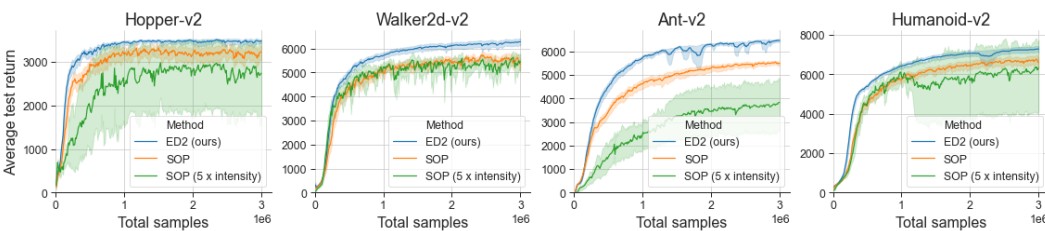

Figure 19: The average test returns across the training of ED2 and SOP with one or five updates for every step in an environment.

## B.2 EXPLORATION

**Vote policy** In this experiment, we used the so-called "vote policy" described in Algorithm 2. We use it for action selection in step 5 of Algorithm 3 in two variations: (1) where the random critic, chosen for the duration of one episode, evaluates each actor's action or (2) with the full ensemble of critics for actors actions evaluation. Figure 20 shows that the arbitrary critic is not much different from our method. However, in the case of the ensemble critic, we observe a significant performance drop suggesting deficient exploration.

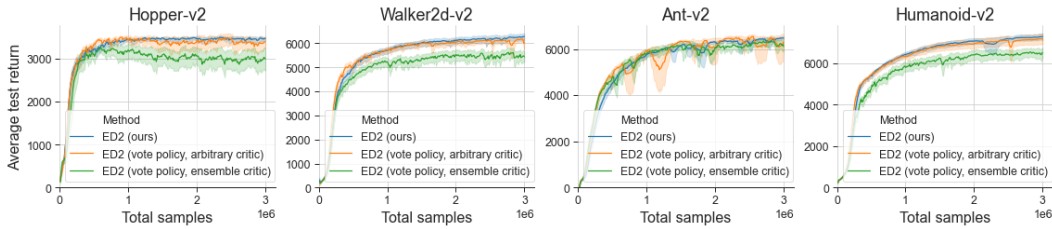

Figure 20: The average test returns across the training of ED2 with and without the vote policy for exploration.

**UCB** We tested the UCB exploration method from Lee et al. (2020). This method defines an upper-confidence bound (UCB) based on the mean and variance of Q-functions in an ensemble and selects actions with the highest UCB for efficient exploration. Figure 21 shows that the UCB exploration method makes the results of our algorithm worse.

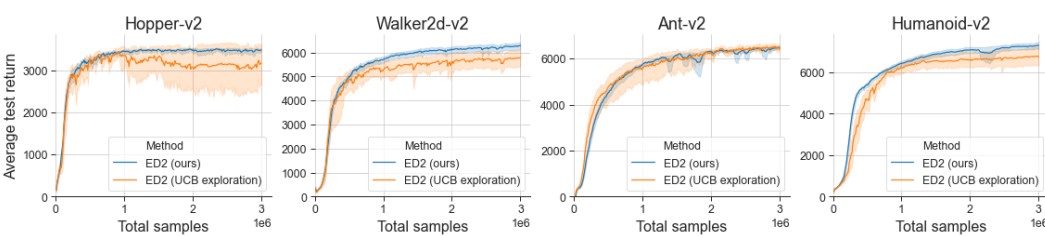

Figure 21: The average test returns across the training of our method and ED2 with the UCB exploration.

**Gaussian noise** While our method uses ensemble-based temporally coherent exploration, the most popular choice of exploration is injecting i.i.d. noise (Fujimoto et al., 2018; Wang et al., 2020). We evaluate if these two approaches can be used together. We used Gaussian noise with the standard deviation of 0.29, it is the default value in Wang et al. (2020). We found that the effects are task-specific, barely visible for Hopper and Walker, positive in the case of Humanoid, and negative for Ant – see Figure 22. In a more refined experiment, we varied the noise level. With more noise the Humanoid results are better, whereas the And results are worse – see Figure 23.

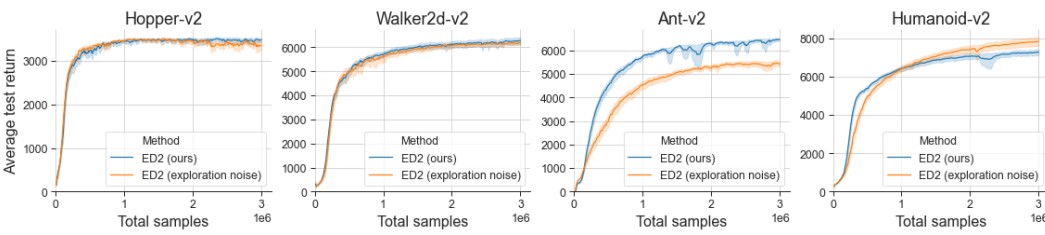

Figure 22: The average test returns across the training of ED2 with and without the additive Gaussian noise for exploration.

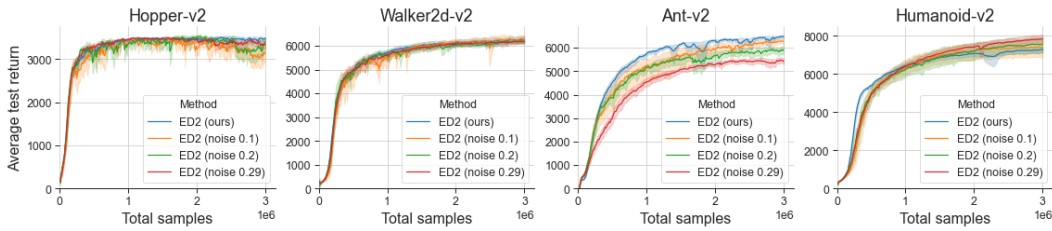

Figure 23: The average test returns across the training of ED2 with and without the additive Gaussian noise for exploration. Different noise standard deviations.

## B.3 EXPLOITATION

We used the vote policy, see Algorithm 2, as the evaluation policy in step 21 of Algorithm 3. Figure 24 shows that the vote policy does worse on the OpenAI Gym MuJoCo tasks. However, on our custom Humanoid tasks introduced in Section 4, it improves our agent performance – see Figure 25.

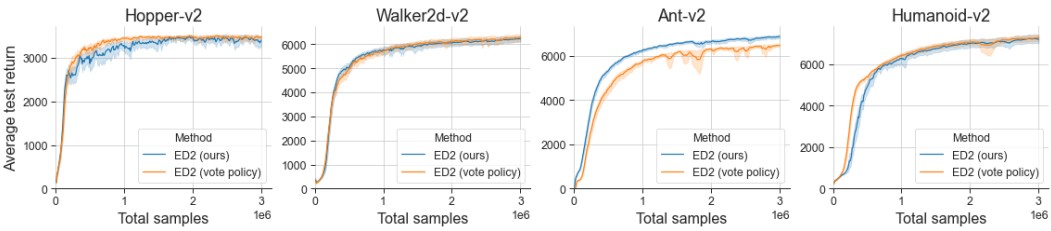

Figure 24: The average test returns across the training of our method and ED2 with the vote policy for evaluation.

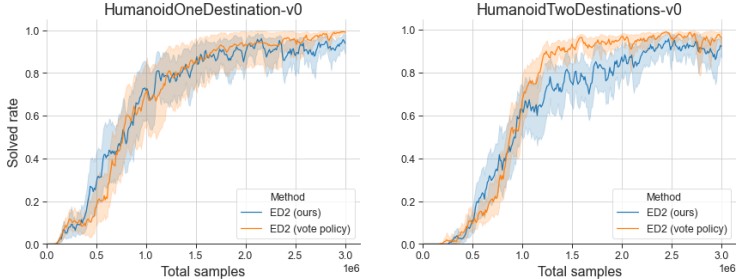

Figure 25: The average test returns across the training of our method and ED2 with the vote policy for evaluation.

## B.4 NORMALIZATION

We validated if rewards or observations normalization (Andrychowicz et al., 2020a) help our method. In both cases, we keep the empirical mean and standard deviation of each reward/observation coordinate, based on all rewards/observations seen so far, and normalize rewards/observations by subtracting the empirical mean and dividing by the standard deviation. It turned out that only the observations normalization significantly helps the agent on Humanoid, see Figures 26 and 27. The action normalization influence is tested in Appendix C.

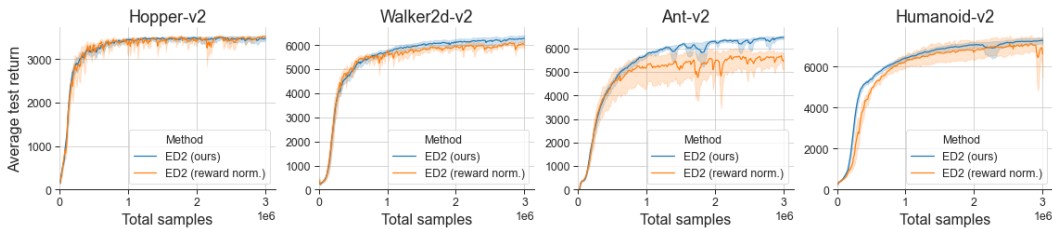

Figure 26: The average test returns across the training of our method and ED2 with the rewards normalization.

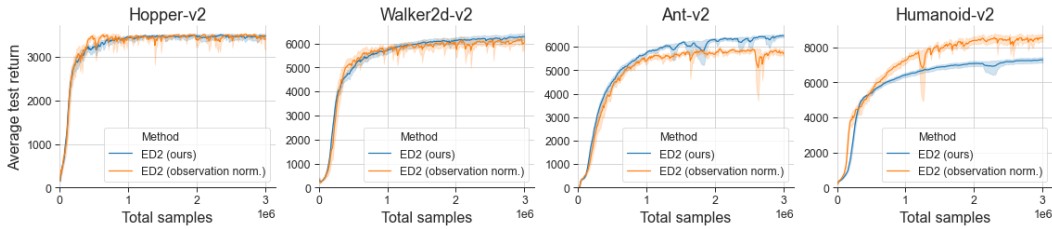

Figure 27: The average test returns across the training of our method and ED2 with the observations normalization.

## B.5 $Q$-FUNCTION UPDATES

**Huber loss** We tried using the Huber loss for the $Q$-function training. It makes the results on all tasks worse, see Figure 28.

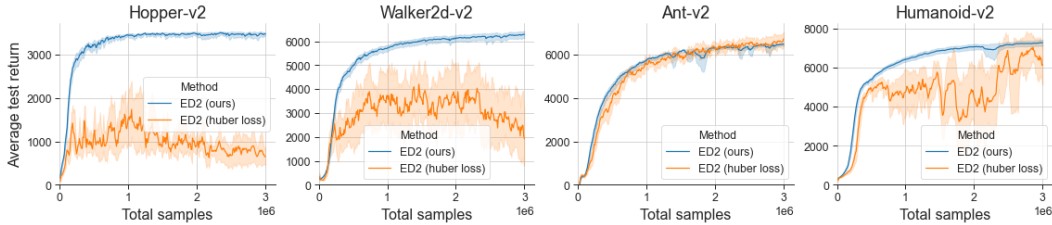

Figure 28: The average test returns across the training of our method and ED2 with the Huber loss.

## C ABLATION STUDY

In this section, we ablate the ED2 components to see their impact on performance and stability. We start with the ensemble exploration and exploitation and then move on to the action normalization and the ERE replay buffer. In all plots, a solid line and a shaded region represent an average and a 95% bootstrap confidence interval over 30 seeds in all but action normalization and ERE replay buffer experiments, where we run 7 seeds.

**Exploration & Exploitation** In the first experiment we wanted to isolate the effect of ensemble-based temporally coherent exploration on the performance and stability of ED2. Figures 29-32 compare the performance and stability of ED2 and one baseline, SOP, to ED2 with the single actor (the first one) used for evaluation in step 21 of Algorithm 3. It is worth noting that the action selection during the data collection, step 5 in Algorithm 3, is left unchanged – the ensemble of actors is used for exploration and each actor is trained on all the data. This should isolate the effect of exploration on

the test performance of every actor. The results show that the performance improvement and stability of ED2 does not come solely from the efficient exploration. ED2 ablation performs comparably to the baseline and is even less stable.

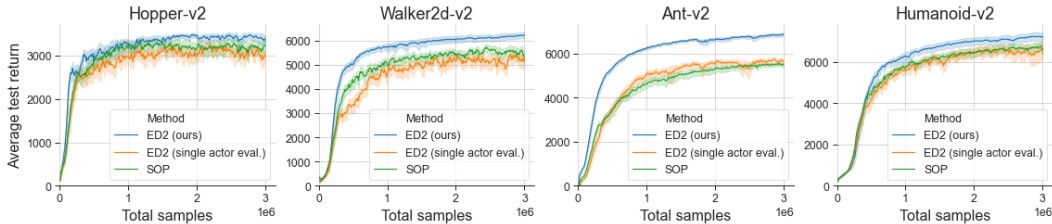

Figure 29: The average test returns across the training of ED2, ED2 with the single actor for exploitation, and the baseline.

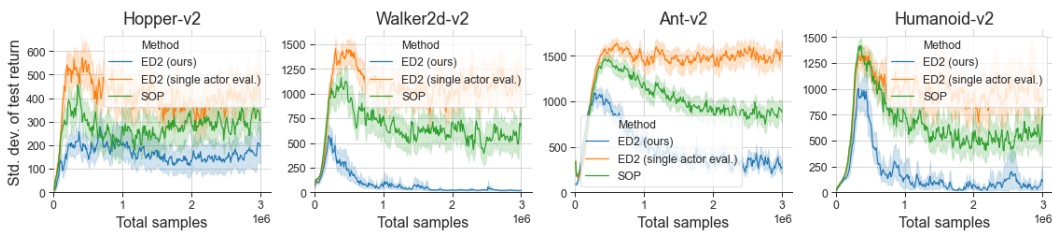

Figure 30: The standard deviation of test returns across the training of ED2, ED2 with the single actor for exploitation, and the baseline.

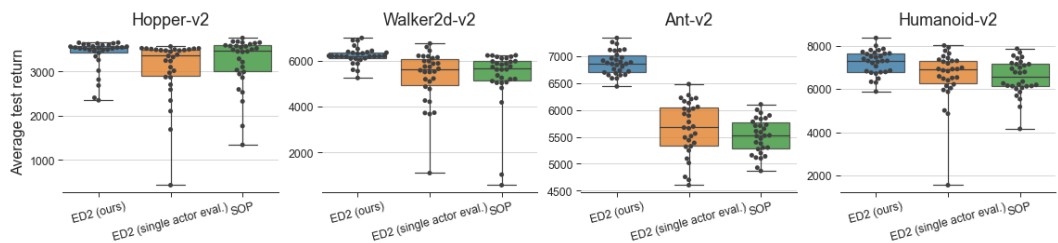

Figure 31: The dots are the average test returns after training ($3M$ samples) of each seed of ED2, ED2 with the single actor for exploitation, and the baseline. The distance between each box's top and bottom edges is the interquartile range (IQR). The whiskers spread across all values.

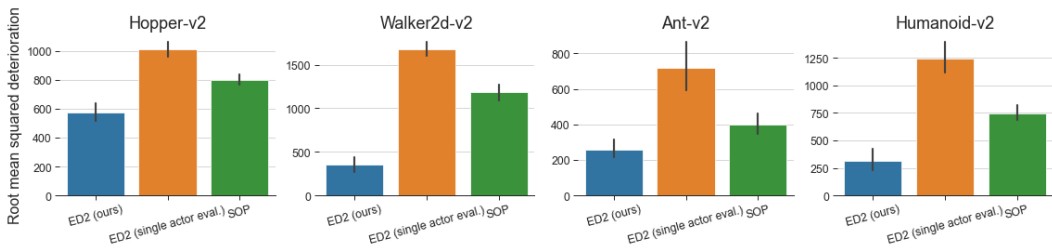

Figure 32: RMSD of ED2, ED2 with the single actor for exploitation, and the baseline – the average and the 95% bootstrap confidence interval over 30 seeds.

In the next experiment, we wanted to check if the ensemble evaluation is all we need in that event. Figure 33 compares the performance of ED2 and one baseline, SOP, to ED2 with the single actor (the first one) used for the data collection in step 5 of Algorithm 3. The action selection during the evaluation, step 21 in Algorithm 3, is left unchanged – the ensemble of actors is trained on the data collected only by one of the actors. We add Gaussian noise to the single actor's actions for exploration as described in Appendix B.2. The results show that the ensemble actor test performance collapses, possibly because of training on the out of distribution data. This implies that the ensemble of actors, used for evaluation, improves the test performance and stability. However, it is required that the same ensemble of actors is also used for exploration, during the data collection.

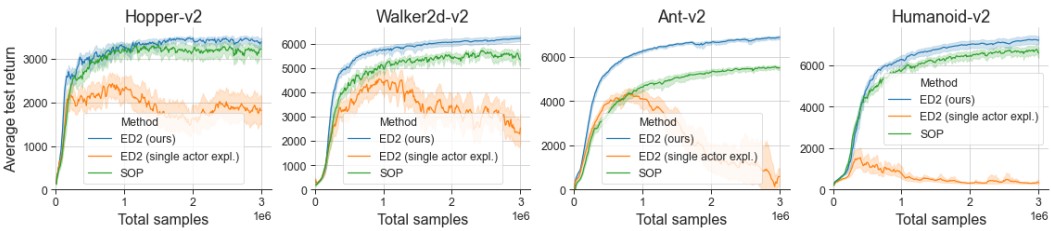

Figure 33: The average test returns across the training of ED2, ED2 with the single actor for exploration, and the baseline.

**Action normalization** The implementation details of the action normalization are described in Appendix E. Figure 34 shows that the action normalization is especially required on the Ant and Humanoid environments, while not disrupting the training on the other tasks.

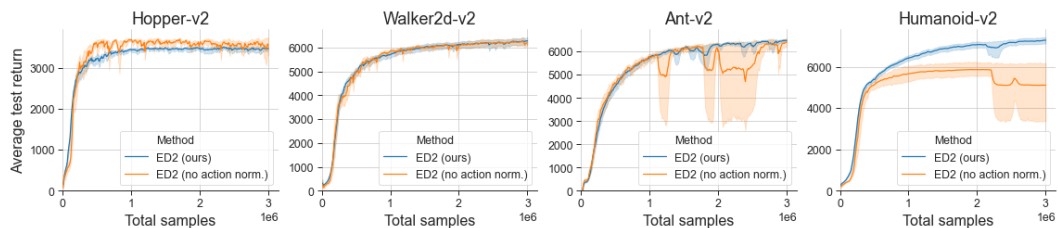

Figure 34: The average test returns across the training of ED2 with and without the action normalization.

**ERE replay buffer** The implementation details of the ERE replay buffer are described in Appendix E. In Figure 35 we observe that it improves the final performance of ED2 on all tasks, especially on Walker2d and Humanoid.

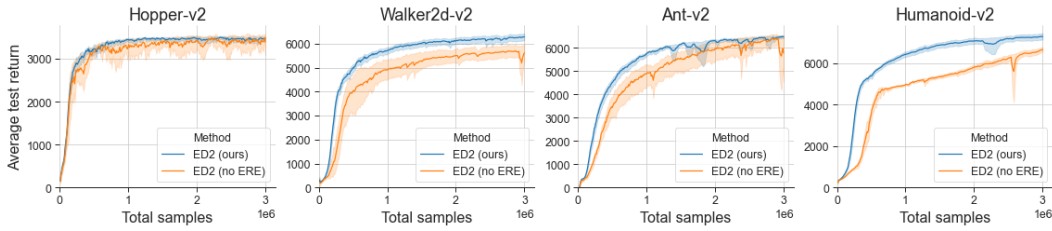

Figure 35: The average test returns across the training of ED2 with and without the ERE replay buffer.

## D   EXPERIMENTAL SETUP

**Plots**   In all evaluations, we used 30 evaluation episodes to better access the average performance of each policy, as described in Section 2. For a more pleasant look and easier visual assessment, we smoothed the lines using an exponential moving average with a smoothing factor equal $0.4$.

**OpenAI Gym MuJoCo**   In MuJoCo environments, that we used, a state is defined by $(x, y, z)$ position and velocity of the robot's root, and angular position and velocity of each of its joints. The observation holds almost all information from the state except the x and y position of the robot's root. The action is a torque that should be applied to each joint of the robot. Sizes of those spaces for each environment are summarised in Table 2.

MuJoCo is a deterministic physics engine thus all simulations conducted inside it are deterministic. This includes simulations of our environments. However, to simplify the process of data gathering and to counteract over-fitting the authors of OpenAI Gym decided to introduce some stochasticity. Each episode starts from a slightly different state - initial positions and velocities are perturbed with random noise (uniform or normal depending on the particular environment).

| Environment name | Action space size | Observation space size |
|---|---|---|
| Hopper-v2 | 3 | 11 |
| Walker2d-v2 | 6 | 17 |
| Ant-v2 | 8 | 111 |
| Humanoid-v2 | 17 | 376 |

Table 2: Action and observation space sizes for used environments.

## E   IMPLEMENTATION DETAILS

**Architecture and hyper-parameters**   In our experiments, we use deep neural networks with two hidden layers, each of them with 256 units. All of the networks use ReLU as an activation, except on the final output layer, where the activation used varies depending on the model: critic networks use no activation, while actor networks use $tanh()$ multiplied by the max action scale. Table 3 shows the hyper-parameters used for the tested algorithms.

| Parameter | SAC | SOP | SUNRISE | ED2 |
|---|---|---|---|---|
| discounting $\gamma$ | 0.99 | 0.99 | 0.99 | 0.99 |
| optimizer | Adam | Adam | Adam | Adam |
| learning rate | $10^{-3}$ | $10^{-4}$ | $10^{-3}$ | $10^{-4}$ |
| replay buffer size | $10^6$ | $10^6$ | $10^6$ | $10^6$ |
| batch size | 256 | 256 | 256 | 256 |
| ensemble size | - | - | 5 | 5 |
| entropy coefficient $\alpha$ | 0.2 | - | 0.2 | - |
| update interval [1] | 50 | 50 | 50 | 50 |
| $\eta_0$ (ERE) | - | 0.995 | - | 0.995 |

[1] Number of environment interactions between updates.

Table 3: Default values of hyper-parameters were used in our experiments.

**Action normalization**   Our algorithm employs action normalization proposed by Wang et al. (2020). It means that before applying the squashing function (e.g. $tanh()$), the outputs of each actor network are normalized in the following way: let $\mu = (\mu_1, \dots, \mu_A)$ be the output of the actor's network and let $G = \sum_{i=1}^{A} |\mu_i|/A$ be the average magnitude of this output, where $A$ is the action's dimensionality. If $G > 1$ then we normalize the output by setting $\mu_i$ to $\mu_i/G$ for all $i = 1, \dots, A$. Otherwise, we leave the output unchanged. Each actor's outputs are normalized independently from other actors in the ensemble.

---

**Algorithm 3** ED2 - Ensemble Deep Deterministic Policy Gradients

---

**Input:** ensemble size $K$; init. policy $\theta_k$ and $Q$-functions $\phi_{k,1}, \phi_{k,2}$ param. where $k \in [1, \ldots, K]$; replay buffer $\mathcal{D}$; max action scale $M$; target smoothing std. dev. $\sigma$; interpolation factor $\rho$;

1: Set the target parameters $\bar{\phi}_{k,1} \leftarrow \phi_{k,1}, \bar{\phi}_{k,2} \leftarrow \phi_{k,2}$
2: Sample the current policy index $c \sim \mathcal{U}([1, \ldots, K])$.
3: Reset the environment and observe the state $s$.
4: **repeat**
5:      Execute action $a = M \tanh\left(\mu_{\theta_c}(s)\right)$            $\triangleright$ $\mu$ uses the action normalization
6:      Observe and store $(s, a, r, s', d)$ in the replay buffer $\mathcal{D}$.
7:      Set $s \leftarrow s'$
8:      **if** episode is finished **then**
9:          Reset the environment and observe initial state $s$.
10:         Sample the current policy index $c \sim \mathcal{U}([1, \ldots, K])$.
11:      **if** time to update **then**
12:         **for** as many as steps done in the environment **do**
13:            Sample a batch of transitions $B = \{(s, a, r, s', d)\} \subset \mathcal{D}$      $\triangleright$ uses ERE
14:            Compute targets

$$y_k(r, s', d) = r + \gamma(1 - d) \min_{i=1,2} Q_{\bar{\phi}_{k,i}}(s', a'_k)$$

$$a'_k = M \tanh\left(\mu_{\theta_k}(s') + \epsilon\right), \epsilon \sim \mathcal{N}(0, \sigma)$$

15:            Update the $Q$-functions by one step of gradient descent using

$$\nabla_{\phi_{k,i}} \frac{1}{|B| \cdot K} \sum_{(s,a,r,s',d) \in B} \left(Q_{\phi_{k,i}}(s, a) - y_k(r, s', d)\right)^2 \quad \text{for } i \in \{1, 2\}, k \in [1, \ldots, K]$$

16:            Update the policies by one step of gradient ascent using

$$\nabla_{\theta_k} \frac{1}{|B| \cdot K} \sum_{s \in B} Q_{\phi_{k,1}}(s, \mu_{\theta_k}(s)) \quad \text{for } k \in [1, \ldots, K]$$

17:            Update target parameters with

$$\bar{\phi}_{k,i} \leftarrow \rho \bar{\phi}_{k,i} + (1 - \rho)\phi_{k,i} \quad \text{for } i \in \{1, 2\}, k \in [1, \ldots, K]$$

18:      **if** time to evaluate **then**
19:         **for** specified number of evaluation runs **do**
20:            Reset the environment and observe the state $s$.
21:            Execute policy $a = \frac{1}{K} \sum_{i=1}^{K} M \tanh\left(\mu_{\theta_i}(s)\right)$ until the terminal state.
22:            Record and log the return.
23: **until** convergence

---

**Emphasizing Recent Experience**    We implement the Emphasizing Recent Experience (ERE) mechanism from Wang et al. (2020). ERE samples non-uniformly from the most recent experiences stored in the replay buffer. Let $B$ be the number of mini-batch updates and $|\mathcal{D}|$ be the size of the replay buffer. When performing the gradient updates, we sample from the most recent $c_b$ data points stored in the replay buffer, where $c_b = |\mathcal{D}| \cdot \eta^{b\frac{1000}{B}}$ for $b = 1, \ldots, B$.

The hyper-parameter $\eta$ starts off with a set value of $\eta_0$ and is later adapted based on the improvements in the agent training performance. Let $I_{recent}$ be the improvement in terms of training episode returns made over the last $|\mathcal{D}|/2$ time-steps and $I_{max}$ be the maximum of such improvements over the course of the training. We adapt $\eta$ according to the formula:

$$\eta = \eta_0 \cdot \frac{I_{recent}}{I_{max}} + 1 - \frac{I_{recent}}{I_{max}}$$

Our implementation uses the exponentially weighted moving average to store the value of $I_{recent}$. More concretely, we define $I_{recent}$ based on two additional parameters $R_{recent}$ and $R_{prev}$ so that $I_{recent} = R_{recent} - R_{prev}$. Those parameters are then updated whenever we receive a new training episode return $ep\_ret$:

$$R_{recent} = \lambda_{recent} \cdot ep\_ret + (1 - \lambda_{recent}) \cdot R_{recent}$$
$$R_{prev} = \lambda_{prev} \cdot ep\_ret + (1 - \lambda_{prev}) \cdot R_{prev}$$

where $\lambda_{prev} = T/\lfloor \frac{|\mathcal{D}|}{2} \rfloor$, $\lambda_{recent} = 10 \cdot \lambda_{prev}$ and $T$ is the maximum length of an episode.

**Hardware**    During the training of our models, we employ only CPUs using a cluster where each node has 28 available cores of 2.6 GHz, alongside at least 64 GB of memory. The running time of a typical experiment did not exceed 24 hours.

