# OpenReview forum: "Continuous Control With Ensemble Deep Deterministic Policy Gradients"
_ICLR.cc/2022/Conference — ICLR 2022 Submitted_

### Official Review · Reviewer_hF9t · 2021-10-25

**Correctness:** 3
**Technical Novelty And Significance:** 2
**Empirical Novelty And Significance:** 3
**Recommendation:** 6
**Confidence:** 4

**Main Review:**

While this paper seems to show some interesting new results related to ensemble reinforcement learning, there are several issues with this paper in this current shape:

1. The technical innovation of this paper remains largely unclear. It was claimed by the authors in this paper that ED2 brings together existing RL tools in a novel way. However, it is unclear which part of the design of ED2 is truly novel. As far as I am aware, ED2 mainly used existing training techniques and ensemble tricks. A series of experiments were carried out to justify the use of several different tricks in ED2. While the combined use of these tricks might be new in ED2, it is not clear why such a combination is potentially more superior than other possible combinations. Furthermore, since the experiments focus mainly on four benchmark problems, it is questionable whether ED2 can achieve clearly better performance over other ensemble baseline algorithms on a much wider range of reinforcement learning problems. Hence the novelty and technical contribution of this paper may need to be improved.

2. This paper lacks theoretical depth. The experiment results in the paper only revealed some insights. However, no further theoretical analysis was conducted to verify (or at least partially explain) the experimental observations. For example, on page 6, the authors conjectured that good exploration may come more from the critic. While this sounds interesting, it is unclear why the critic will play such a critical role to induce effective exploration and what the corresponding conditions are for this to happen.

3. Some experiment findings and the corresponding claims do not appear to be consistent. For example, on page 4, the authors found experimentally that additive normal action noise can substantially improve the Ant performance. They subsequently concluded that additive noise is not required for effective learning. These two claims do not sound consistent. Accordingly, the main findings discovered in the paper may need to be further verified.

4. Some experiment findings appear to be well-known a priori in the literature. For example, as acknowledged by the authors, posterior sampling techniques can be more effective than the OFU strategy for action selection. Consequently, the technical contribution of the corresponding experiment results does not seem to be sufficiently strong.

**Summary Of The Paper:**

This paper conducted an experimental study over a range of tricks that are often exploited to facilitate ensemble deep reinforcement learning. The experiment results show several interesting findings. For example, it was found that commonly used additive action noise may not be necessary for effective exploration. Meanwhile, experiments show that the initialization of critics perhaps has a higher impact on learning performance than the initialization methods adopted for actors. These findings can be quite important to guide future design of more effective ensemble reinforce learning algorithms.

**Summary Of The Review:**

This paper conducted an experimental study over a range of tricks that are often exploited to facilitate ensemble deep reinforcement learning. The corresponding empirical findings can be quite important to guide future design of more effective ensemble reinforce learning algorithms. Meanwhile, the technical novelty and theoretical depth of this paper may need to be strengthened.

---

> ### Author Response · Authors · 2021-11-22
> **We kindly thank the reviewer for their recognition and the comments. We address the reviewer comments in order**
>
> We kindly thank the reviewer for their recognition and the comments. We address the reviewer comments in order.
>
> Comment #1
>
> For a fair comparison, we run additional experiments where we compare ED2 to the ensemble of SAC and SOP runs. Because of the short space of time, we present performance results for only some variants and with a limited number of seeds (at least 3 and up to 10 for each curve). However, they still present the effects we want to discuss. We will run the complete study comparing ED2 performance & stability to the ensemble SAC and SOP variants for the camera-ready paper.
>
> In the figures below, we present the ensemble of SAC runs performance (ESAC) and the ensemble of SOP runs performance (ESOP), where the actors are combined either by averaging the policies (mean policy) or by using voting described in the paper (vote policy), and compare them ED2 and non-ensemble SAC and SOP – conclusions are:
>
> * The voting policy in general does better than the mean policy. We hypothesise that because ESAC and ESOP train their agents without the shared replay buffer, the agents converge to different behaviour and averaging the policies make these different behaviours compete with each other. In the vote policy case, the ensemble of critics decides on only one best action from the five proposed by actors and this effect is not visible so much. We plan to test this hypothesis for the camera-ready paper.
> * Only ESAC mean policy and ESOP vote policy achieve the near ED2 performance on Hopper. Any other method, in any other environment, does worse or much worse than ED2.
> * Moreover, the ensemble versions of SAC and SOP often do worse even than the non-ensemble agents.
>
> [ensembles-test.png](https://postimg.cc/YGT8jZYz)
>
> [ensemble-vs-sac.png](https://postimg.cc/qtb1Hz7Y)
>
> [ensemble-vs-sop.png](https://postimg.cc/sQMwjsQ2)
>
> To support our results, we present the average online performance of the separate actors of each ensemble agent – i.e. for each ensemble agent it is the average curve of the data collection performance of each actor in its ensemble. The figure below shows that each actor successfully trains in separation and the performance only breaks when the actors get combined into their ensembles.
>
> [ensembles-train.png](https://postimg.cc/bSbgngRc)
>
> This experiment supports the novelty of the ED2 algorithm as it is not enough to simply combine independent runs of SAC or SOP – neither by averaging policies nor by using voting. Our technical innovation is in showing how to successfully construct a state-of-the-art performing and stable algorithm using a straightforward setup and a relatively small number of RL tricks. In doing so, we tested a variety of setups, providing insight into what works and what does not, which is hard to come by. We argue that this work is valuable both for practitioners and researchers, and allows them to build on top of our work while avoiding pitfalls that we described and tested empirically.
>
> We have run extensive experiments of our design choices on the MuJoCo environments as the representation of the continuous control domain. Repeating all of these experiments for other environments, like DM Control Suite, is a major endeavour and is postponed for future work.
>
> Comment #2
>
> Our paper fits into the empirical body of RL research, which is explicitly emphasized in Abstract and Introduction. The scope of experiments is extensive – we study 25 phenomena, implement 4 algorithms in many variants, and use 30 seeds. While theoretical analysis would most certainly enrich our understanding of RL, the inclusion of additional theoretical material would result in the removal of substantial parts of the paper, hurting the scope of analysis and risking the loss of a focus. However, we do hope that our work opens the possibility of building explanations and further improvements in future work. For example, internally, we believe that it is bootstrap learning which lets critics maintain their diversity and thus drive exploration. We found this hypothesis too speculative to include at the present moment.
>
> Comment #3
>
> The reviewer may have understood that the action noise improved the Ant performance, whereas the lack of it improves the Ant performance as can be seen in Figure 6 of the revised text. When viewed this way, the results are consistent (the additive noise is not required for effective learning). We rephrased the appropriate sentence in the revised text.
>
> Comment #4
>
> Our technical contribution is the empirical evaluation of the heuristic algorithms. We prove empirically that the theoretical superiority of the posterior sampling over the OFU strategy holds for the approximated case of ED2 and SUNRISE in the continuous domains, which was not known up to our best knowledge. We clarified it in the revised text of Section 4.4.

---

> > ### Comment · Reviewer_hF9t · 2021-11-28
> > **Thank the authors to respond to my comments.**
> >
> > I would like to thank the authors to respond to my comments. The response has addressed some of my concerns. However, I still have concern that the experimented problems may be limited for the purpose of "empirically proving" the effectiveness of the new algorithm design. While the new design may work well on some MuJoCo problems, there is no guarantee that the same design can work well consistently on other reinforcement learning problems. Due to this reason, some theoretical analysis appears essential so that I can clearly see why certain algorithm design can improve learning performance across a wide range of different reinforcement learning problems. Hence, while the paper opens room for future research, I am not 100% confident about the conclusions drawn in the paper. For a paper focusing on empirical research, this appears to be a limitation to me.

---

### Official Review · Reviewer_M5Fw · 2021-11-01

**Correctness:** 3
**Technical Novelty And Significance:** 3
**Empirical Novelty And Significance:** 3
**Recommendation:** 6
**Confidence:** 5

**Main Review:**

Strength:

The main strength of the work is introducing a straightforward extension of an existing base actor-critic method that substantially outperforms existing algorithms in standard benchmark tasks. The new algorithm also has some desirable properties for policy optimization such as not having random additive noise and providing stable performance.

Moreover, some key insights on deep policy gradient methods are presented such as the contribution of actor and critic initialization.

Another strength of the paper is its focus on various details such as ideas that didn’t work as well as ablative studies in various manners.

Weakness:

The main weakness of the work is the fairness of the experiments. This deficiency is common in many papers including those that get published in top conferences but acceptance doesn’t justify wrong choices. I would be willing to know from the authors their thoughts on it.

The first issue with such experiments is that only a single hyper-parameters are used for the methods to show comparison. However, the claims concluded from such results are that one algorithm outperformed the others. However, to make such a claim, different hyper-parameter values should be tried for all algorithms. Otherwise, the claim should be humbler such as our method outperforms the competing methods with default choices of hyper-parameters and such. Such hyper-parameter search is also necessary for ablative studies. When we are removing one component at a time, we cannot assume that the default hyper-parameter configuration of the original method will still be effective for the subsequent variants.

I understand that it will make deep RL experiments much more expensive. However, it isn’t necessary to perform a grid search over hyper-parameters. It has been shown before that random search can give close to the best performance within a handful of trials of configurations, which will considerably reduce the search cost.

In a similar vein, it has been common to compare algorithms with different computational profiles. However, when a new algorithm is computationally way more expensive than the competitors, is it fair to compare them with such computational disparity and claim one algorithm is better than the other?

Details on the computational expense of ED2 are not given. How many actors are used for ED2? How much more expensive ED2 is compared to its competitors such as SAC or SOP?

Other comments:
- To understand more clearly, det. SOP uses no exploration whatsoever and still performs well on these tasks?

- Both the action averaging and greedy choice among actors yielded similar results. This leads me to suspect whether the actors either converged to similar performant behavior or stationary behavior with zero torque, which upon averaging gives a behavior similar to the greedy one.

- Considering the performance and the computational expense compared to ED2, det. SOP seems a strong contender. Why is it not added to Figures 6, 7, or 8?

- Figure 9 somewhat makes sense except that there is a puzzle. When ED2 is reduced to single critic, the diversity is reduced considerably, which hurts possibly the exploration and reduces performance substantially. But if we reduce ED2 single critic further by also having a single actor, then don’t we get det. SOP, which wasn’t doing as badly as ED2 single critic? How can that be explained?

- What's really the motivation behind having a separate evaluation phase just to measure performance for plot when learning online? If these algorithms are deployed to learn online say on a robot, their online performance is the actual evaluation. Creating an additional evaluation phase to measure performance will only delay its learning in real-time. In what case, such a separate evaluation is useful other than because many other works repeat it? Even if there is a case, isn't it quite restrictive? Wouldn't it be important to see the online performance of ED2 as it randomly draws actors to interact? If that performance is also good, it would be a more interesting and stronger result.

**Summary Of The Paper:**

This paper has two main contributions: it introduces an ensemble-based actor-critic method, and it answers some pertinent questions in policy optimization by focusing on its different components. The ensemble is different from multi-actor learners that interact with multiple environments simultaneously, violating the standard RL setup. Instead, the learner of this paper maintains multiple actors and critics but uses only a single actor at a time to interact with the environment. All actors and critics are trained on a common replay buffer. The base method is the streamlined off-policy (SOP) method, which unlike soft actor-critic (SAC) doesn’t use an entropy bonus. Additionally, no exploration noise is added, resulting in their Ensemble Deep Deterministic (ED2) method.

The proposed algorithm ED2 is shown to be superior and more stable in performance according to different measures compared to existing methods. It is also revealed that actor initialization affects performance less than critic initialization. ED2 uses deterministic actors, and its exploration comes from sampling among the actors. Such a form of exploration is also shown to be superior to UCB-style exploration.


**Summary Of The Review:**

The paper’s strength is a straightforward performant policy optimization method and the insights developed through experiments. However, hyper-parameter search isn’t performed to substantiate the strong claims and it isn’t clear how much more computationally expensive ED2 is compared to its competitors.

*** updated ***

I will update later.

---

> ### Author Response · Authors · 2021-11-22
> **We kindly thank the reviewer for their recognition and the comments**
>
> We kindly thank the reviewer for their recognition and the comments.
>
> As the reviewer already noted, we ran extensive experiments of our design choices and repeating all of these experiments for different hyper-parameters choices, even if using random search, is a major endeavour and is postponed for future work. However, we stress that the default hyper-parameters were fine-tuned with respect to their methods. For our method, we fine-tuned the parameters we introduce or test (i.e. ensemble size or normal noise std. dev.), which we believe makes our results significant.
>
> For a fair comparison, we run additional experiments where we compare ED2 to the ensemble of SAC and SOP runs. Because of the short space of time, we present performance results for only some variants and with a limited number of seeds (at least 3 and up to 10 for each curve). However, they still present the effects we want to discuss. We will run the complete study comparing ED2 performance & stability to the ensemble SAC and SOP variants for the camera-ready paper.
>
> In the figures below, we present the ensemble of SAC runs performance (ESAC) and the ensemble of SOP runs performance (ESOP), where the actors are combined either by averaging the policies (mean policy) or by using voting described in the paper (vote policy), and compare them ED2 and non-ensemble SAC and SOP – conclusions are:
>
> * The voting policy in general does better than the mean policy. We hypothesise that because ESAC and ESOP train their agents without the shared replay buffer, the agents converge to different behaviour and averaging the policies make these different behaviours compete with each other. In the vote policy case, the ensemble of critics decides on only one best action from the five proposed by actors and this effect is not visible so much. We plan to test this hypothesis for the camera-ready paper.
> * Only ESAC mean policy and ESOP vote policy achieve the near ED2 performance on Hopper. Any other method, in any other environment, does worse or much worse than ED2.
> * Moreover, the ensemble versions of SAC and SOP often do worse even than the non-ensemble agents.
>
> [ensembles-test.png](https://postimg.cc/YGT8jZYz)
>
> [ensemble-vs-sac.png](https://postimg.cc/qtb1Hz7Y)
>
> [ensemble-vs-sop.png](https://postimg.cc/sQMwjsQ2)
>
> To support our results, we present the average online performance of the separate actors of each ensemble agent – i.e. for each ensemble agent it is the average curve of the data collection performance of each actor in its ensemble. The figure below shows that each actor successfully trains in separation and the performance only breaks when the actors get combined into their ensembles.
>
> [ensembles-train.png](https://postimg.cc/bSbgngRc)
>
> This experiment supports the novelty of the ED2 algorithm as it is not enough to simply combine independent runs of SAC or SOP – neither by averaging policies nor by using voting. Our technical innovation is in showing how to successfully construct a state-of-the-art performing and stable algorithm using a straightforward setup and a relatively small number of RL tricks. In doing so, we tested a variety of setups, providing insight into what works and what does not, which is hard to come by. We argue that this work is valuable both for practitioners and researchers, and allows them to build on top of our work while avoiding pitfalls that we described and tested empirically.
>
> Considering the compute expense, there are 5 actors and 5 critics used in the ED2 algorithm. This is noted in Section 3 and the ensemble size is fine-tuned in Section 4.1 in the revised text.
> We run computations on 28 core CPU nodes with 64GB of RAM. The training time of the non-ensemble SOP agent vs the ensemble ED2 takes around 6h, 6h, 8h, and 10h vs. 9h:45m, 10h, 12h:45m, and 17h on Hopper, Walker, Ant, and Humanoid respectively. This means that ED2 takes around +65% time to train (while encompassing 5x amount of networks/weights).
> We shall add that ED2 not only achieves better performance than the state-of-the-art methods, but it achieves their state-of-the-art performance with fewer samples e.g. SUNRISE requires around 1,75M samples to get to the score of 6000 on Ant, whereas ED2 requires only around 0,75M samples. It is 1M samples (more than 2 times) fewer. Moreover, SOP and SAC require 3M samples to get to the score of 5500 on Ant. ED2 requires only around 0,5M samples, which is 6x fewer samples. ED2 is also the same or more sample-efficient in the rest of the environments.
>
> We address other comments in the next response.

---

> > ### Author Response · Authors · 2021-11-22
> > **We address other comments in order**
> >
> > We address other comments in order
> >
> > Comment #1
> >
> > Deterministic SOP uses no exploration noise or any other additional exploration mechanism for data collection and it still performs well on MuJoCo tasks.
> >
> > Comment #2
> >
> > Thank you for your suggestion of the research question (whether the actors either converged to similar performant behaviour or stationary behaviour with zero torque). We planned to verify this and we may be able to do this for the camera-ready paper.
> >
> > Comment #3
> >
> > We present the stability results for the det. SOP. Besides its better stability across evaluation runs and training seeds on Hopper, it is similar to SOP, which supports our claim that the normal action noise is not required for effective training. We will add this result to Appendix, as putting it alongside SUNRISE and SAC clutters the figures.
> >
> > [ed2-sop-sac-det-sop-std-dev.png](https://postimg.cc/21g1sGxZ)
> >
> > [ed2-sop-sac-det-sop-train-stability.png](https://postimg.cc/yDDJpt0K)
> >
> > [ed2-sop-sac-det-sop-rmsd.png](https://postimg.cc/D8jJgnJ5)
> >
> > Comment #4
> >
> > We have rerun the ED2 experiment with the single critic and indeed there was some mistake in the original plot. It is still clear that the single critic ED2 version does worse, but the results are closer to (or the same as) the deterministic SOP which is expected as the reviewer noted.
> >
> > [ed2-single-critic-det-sop.png](https://postimg.cc/1VwtZVyr)
> >
> > Comment #5
> >
> > Indeed online performance is an important metric. However, during evaluation, we focus on exploitation, which (in practice) means we use the whole ensemble. We also average more evaluation runs to make the result variance lower and hence plots smoother. It also allows us to isolate the training variance on the plots with CI’s across seeds. We note that we study the evaluation variance separately in Section 4.2.
> > We include the comparison between the online and the evaluation performance of the algorithms from the paper. As expected, the performances are the same for the det. SOP (which uses the same policy during evaluation and for data collection) and the evaluation performance is often better for all other methods which disable their exploration mechanisms for evaluation (e.g. in SAC we don’t sample actions, but take the mean returned by the neural network).
> >
> > [ed2-train-vs-eval.png](https://postimg.cc/bD2YQSSf)
> >
> > [sunrise-train-vs-eval.png](https://postimg.cc/y3mVGqTf)
> >
> > [sac-train-vs-eval.png](https://postimg.cc/3yqJ4FRZ)
> >
> > [sop-train-vs-eval.png](https://postimg.cc/m1fLVZ7z)
> >
> > [det-sop-train-vs-eval.png](https://postimg.cc/R3hZZDMF)

---

### Official Review · Reviewer_ZQ4d · 2021-11-02

**Correctness:** 3
**Technical Novelty And Significance:** 2
**Empirical Novelty And Significance:** 2
**Recommendation:** 5
**Confidence:** 4

**Main Review:**

Pros:
- The code is open-sourced, which is excellent for reproducibility
- There is significant focus on the stability of RL algorithms, which is great to see. Often RL papers just present the reward curves and information on the stability of these algorithms is important.
- Using 30 random seeds (instead of the usual 3-6) is great and strengthens any empirical claims

Cons:
- The empirical results don’t seem to justify the algorithm. Figure 4 represents the main empirical argument for ED2 but the empirical gains shown are minor. On Walker it appears as though SUNRISE actually outperforms ED2. Humanoid and Hopper both have ED2 on top, but only by a very slim margin (one that has overlapping confidence intervals with other algorithms).
- Small number of environments. The algorithms are only compared on 4 environments (Ant, Hopper, Humanoid, Walker) when MuJoCo/OpenAI Gym offers many more (e.g. HalfCheetah, InvertedDoublePendulum, Swimmer, Reacher, etc.). The empirical results could be more compelling if the scope was increased. If you expanded the results presented in Table 1 for all algorithms, this could be a more compelling argument.
- Much of the paper is dedicated to information that would be better in the appendix. Information about non-essential ablation studies (e.g. Figure 2 shows the impacts are pretty minor) are not necessary in the main body. These figures could be replaced with more directly relevant information, like to aid claims of stability, e.g. by injecting varying degrees of noise/randomness into the environment and evaluating relative performances, or evaluating sensitivity to other hyperparameters (network structure, learning rates, etc.)
- Lack of theoretical understanding. I am well aware of the problems of equation packing and the disconnect that often occurs in RL papers between the algorithm and the theoretical justification, but at least providing some insight into comparisons of UCB, posterior sampling, priors, etc. would be nice.
- The empirical comparisons don’t seem to offer an even playing field for all algorithms. Based on (Haarnoja et al., 2019), SAC should reach ~8000 on Humanoid-v2 (after 10e6 steps). However, the experiments are only conducted till 3e6 steps with multiple algorithms appearing to still have upward trajectories. There seems to be no mention of a focus on the speed of learning/early learning performance vs final performance.
- The relationship between Figure 4 and Figure 6 is not clear. Figure 6 seems to indicate that (e.g.) pretty much all algorithms have >1,000 STD on Humanoid and Ant early on in training. However, the bootstrapped CI in Figure 4 is substantially smaller. This discrepancy is uncommon and should be explained.
- Reinforcement learning is notoriously brittle and I would encourage references to the previous work done evaluating the effect of random seeds/initialization on agent performance e.g. the famous (Henderson et al., 2019).
- Figure 5 does not provide any support or meaningful insight. The velocity of humanoid is not what the algorithm is optimizing. It is included for “completeness”, but this seems to be a cherry-picked statistic that doesn’t convey anything meaningful (while the speed is incorporated in the reward function, since the rewards are much closer than the velocities we don’t see what the tradeoff is).
- Figure 1 does not really help clarify anything. If one already knows the environments, then the figure is unnecessary, if one is unfamiliar with them, the figure doesn’t show what actually transpires in the environment and doesn’t clear anything up.
- To claim ED2 is really SotA, further analysis is necessary across environments like Agarwal et al. 2021, Barreto et al. 2010, Jordan et al. 2020, etc.
- There are minor typographical inconsistencies, e.g. differing usages of \cite{} and \citep{}
- Inconsistent background information. The paper provides a definition of standard deviation, an extremely common statistical measurement, but not for much more niche terms such as approximated UCB. To be clear, I have no problem with giving the formula for STD, but not giving definitions for much less widely known terms is something that could be fixed.

Misc/Note:
This paper seems like it’s trying to do two things at once: (1) provide a review of RL techniques (e.g. exploitation techniques), evaluate their impact and report on the key takeaways of this empirical analysis, and (2) introduce a novel RL algorithm and justify it’s empirical construction and performance. Both of these are papers that are perfectly fine, but by trying to do both it leaves something lacking from each of them. If this was a review of techniques, I would like to see more continuous control algorithms evaluated and more techniques experimented with. If this is just introducing a novel algorithm, I would like to see more theoretical explanations of the techniques (e.g. theoretical derivations and insights into the effect of K)  and more extensive ablation studies (e.g. evaluating on more MuJoCo tasks or other continuous control environments that have different properties such as RLBench, Industrial control benchmark, assistive gym, DM Control suite, etc.).


**Summary Of The Paper:**

This paper presents a deep reinforcement algorithm, Ensemble Deep Deterministic Policy Gradients (ED2), for continuous control tasks. The algorithm is empirically derived and is claimed to represent SotA performance on several tasks and while providing more stable results. These claims are justified based primarily on the (reward and stability) results on 4 MuJoCo environments.


**Summary Of The Review:**

Given that the algorithm is entirely empirically derived and the empirical results are not compelling, I have given this paper a reject. I appreciate that there are important ideas represented here, but it currently isn’t up to the ICLR standard.

After author rebuttal, the score has moved from 3 to 5 (see comment for more information).

---

> ### Author Response · Authors · 2021-11-22
> **We kindly thank the reviewer for your careful reading of our paper**
>
> We kindly thank the reviewer for your careful reading of our paper. We would like to address the comments in the same order as given.
>
> Comment #1
>
> We respectfully disagree: the 95% bootstrap confidence intervals do not overlap by a big margin in the Ant and Humanoid environments while being at least on par in other environments.
>
> Comment #2 and #3
>
> We have run extensive experiments of our design choices on the MuJoCo environments, representative of the continuous control domain. Repeating all of these experiments for other environments, like DM Control Suite, is a major endeavour and is postponed for future work.
>
> Comment #4 and #11
>
> We moved more relevant results into the revised text like hyper-parameter search (the ensemble size) and ablation study experiments (ED2 with and without the action noise) to Section 4 and removed less significant figures like the MuJoCo environment plot. We thank you for this suggestion.
>
> Comment #5
>
> We will add a section with appropriate definitions and references to Osband & Van Roy (2017) and Lattimore & Szepesvári (2020), coupled with a short discussion. Having said that, the extensive treatment of this topic is outside of the scope of this paper.
>
> Comment #6
>
> We will enrich the background section of the camera-ready paper with the approximate UCB and bootstrap posterior sampling explanations or at least put them in the appendix (depending on the space constraints).
>
> Comment #7
>
> We focus on the mid-training performance where we find the most interesting effects happen. We believe this is a relevant area for making progress towards data-efficient reinforcement learning. In particular, differences for 10e6 steps get blurry, for e.g., PPO and SAC (fixed temperature) on Humanoid [Haarnoja et al., 2019].
>
> Comment #8
>
> Figures 4 and 6 present different variances: training variance vs. evaluation variance. Training variance is calculated across training runs and each data point is mean across 30 evaluation runs. Evaluation variance shows the variance across these runs. We note that ED2 reduces both the training variance and evaluation variance.
>
> Comment #9
>
> We follow at least some of the Henderson et al. recommendations. We run experiments on 30 seeds and include the Humanoid velocity in order to show the metrics that quantitatively represent learning of the desired behaviour relevant to the environment’s objective (the learning curve can show progress in optimizing a local optimum). We will properly cite this work in the camera-ready paper.
>
> Comment #10
>
> The standard reward function as defined in Mujoco consists of several components. Some of them are designed to improve learning; however, they may obscure the true objective of moving. Reporting velocity is arguably more aligned with this goal. For instance, the return for Humanoid simply standing is close to 5000 (\approx 5 living reward * 1000 time steps). That being said, we understand such presentations can be thought of as “cherry-picked” as we don’t present similar metrics for other envs at the moment. Thus we have decided to remove it from the revised text to save space for other experiments.
>
> Comment #12
>
> We have fixed the typographical inconsistencies (\cite{} vs \citep{}) in the “Introduction” and “Related work” sections in the revised text.
>
> References
>
> [Lattimore & Szepesvári, 2020] Lattimore, T., & Szepesvári, C. (2020). Bandit Algorithms. Cambridge: Cambridge University Press. doi:10.1017/9781108571401
>
> [Osband & Van Roy, 2017] Osband, Ian, and Benjamin Van Roy. ‘Why Is Posterior Sampling Better than Optimism for Reinforcement Learning?’ In Proceedings of the 34th International Conference on Machine Learning, ICML 2017

---

> > ### Comment · Reviewer_ZQ4d · 2021-11-29
> > **Thank the authors for comments, insights, and clarifications**
> >
> > I appreciate the authors responses to my comments. They have helped to alleviate some of the concerns I had, especially with regard to presentation and understanding. To this end, I have modified my suggestion from 3 to 5. However, the core concerns remain. Specifically, that for an empirically derived algorithm there should be more empirical evidence. To this end I agree with reviewer hF9t (in their response to author rebuttal). As you indicated in your response, there are future plans to incorporate more experiments. I believe that those have the potential to make a strong paper (as M5Fw agreed in their top comment), but its current form remains lacking.

---

### Official Review · Reviewer_2e3U · 2021-11-03

**Correctness:** 2
**Technical Novelty And Significance:** 2
**Empirical Novelty And Significance:** 3
**Recommendation:** 3
**Confidence:** 2

**Main Review:**

strengths
1. The paper is also well-written and easy to understand
2. I believe the paper studies a problem that's important and significant in the rl community and very relevant to the venue

weaknesses/questions
In general, I think the results should be treated more carefully and some theoretical motivation is needed. Following are some of my questions.
1.in section 4.1, the conclusion of "This result shows that no additional exploration mechanism, often in a form of an exploration noise (Lillicrap et al., 2016; Fujimoto et al., 2018; Wang et al., 2020), is required for the diverse data collection and it can even hinder training." seems rather strong judging from fig.2 (I would say they are roughly the same).
2. "Figure 3 shows that ED2 magnifies the beneficial effect coming from the deterministic exploration.", wouldn't it make better sense if you compare against the ones that are also ensemble, e.g. ensemble SOP instead of SOP? How do you know what brings the performance, is it the ensemble or the determinist action, or both?
3. fig.4 again, why not compare the ensemble baselines? I think it's important to ablate the design choices that are actually important

Some minor problems.
1. maybe more introduction for SOP (e.g. what's the ere replay buffer?)
2.sec.3, "These two choices ensure coherent and temporally-extended exploration similarly to Osband et al. (2016).", I do not understand why is the exploration "coherent and temporally-extended".
3.in sec.3, I guess the used and not used should be comparable and matched, but why is action normalization compared with "observations and rewards normalization"?
4.fig.5, what does it mean for the Humanoid velocities? could you elaborate?

**Summary Of The Paper:**

This paper presents an empirical study between different commonly-used tricks and elements of off-policy RL algorithms and tries to understand the interplay between those elements. The authors propose a new method called ED2 that utilizes their insights from the empirical study.

**Summary Of The Review:**

I believe the paper should present a more careful analysis of these design elements and put more effort into justifying their choices used for comparison.

---

> ### Author Response · Authors · 2021-11-22
> **We kindly thank the reviewer for the feedback.**
>
> We kindly thank the reviewer for the feedback.
>
> Our paper fits into the empirical body of RL research, which is explicitly emphasized in Abstract and Introduction. The scope of experiments is extensive – we study 25 phenomena, implement 4 algorithms in many variants, and use 30 seeds. While theoretical analysis would most certainly enrich our understanding of RL, the inclusion of additional theoretical material would result in the removal of substantial parts of the paper, hurting the scope of analysis and risking the loss of a focus. However, we do hope that our work opens the possibility of building explanations and further improvements in future work. For example, internally, we believe that it is bootstrap learning which lets critics maintain their diversity and thus drive exploration. We found this hypothesis too speculative to include at the present moment.
>
> We would like to address the reviewer’s comments.
>
> Question #1
>
> Our findings challenge the statement that random dithering is needed for efficient RL exploration. As it is a relatively common belief in RL (almost always used in practice) we decided to underline this fact. We admit that our phrasing is perhaps too strong, and we have rephrased the conclusion of Section 4.3 in the revised text.
>
> Question #2
>
> We advocate for studying the influence of ensembles and deterministic data collection separately. For example, the additional experiments in Appendix, see Figure 22 (or 23 in the revised text), indicate that using deterministic actions with and without the ensemble has similar effects of boosting Ant performance and slightly decreasing the Humanoid performance. In the revised text, in order to improve clarity, we have separated the ensemble of agents experiments from the action noise experiments (see Section 4). We also moved the aforementioned figure and the ensemble size hyper-parameters search results to the new subsections.
>
> Question #3
>
> For a fair comparison, we run additional experiments where we compare ED2 to the ensemble of SAC and SOP runs. Because of the short space of time, we present performance results for only some variants and with a limited number of seeds (at least 3 and up to 10 for each curve). However, they still present the effects we want to discuss. We will run the complete study comparing ED2 performance & stability to the ensemble SAC and SOP variants for the camera-ready paper.
>
> In the figures below, we present the ensemble of SAC runs performance (ESAC) and the ensemble of SOP runs performance (ESOP), where the actors are combined either by averaging the policies (mean policy) or by using voting described in the paper (vote policy), and compare them ED2 and non-ensemble SAC and SOP – conclusions are:
>
> * The voting policy in general does better than the mean policy. We hypothesise that because ESAC and ESOP train their agents without the shared replay buffer, the agents converge to different behaviour and averaging the policies make these different behaviours compete with each other. In the vote policy case, the ensemble of critics decides on only one best action from the five proposed by actors and this effect is not visible so much. We plan to test this hypothesis for the camera-ready paper.
> * Only ESAC mean policy and ESOP vote policy achieve the near ED2 performance on Hopper. Any other method, in any other environment, does worse or much worse than ED2.
> * Moreover, the ensemble versions of SAC and SOP often do worse even than the non-ensemble agents.
>
> [ensembles-test.png](https://postimg.cc/YGT8jZYz)
>
> [ensemble-vs-sac.png](https://postimg.cc/qtb1Hz7Y)
>
> [ensemble-vs-sop.png](https://postimg.cc/sQMwjsQ2)
>
> To support our results, we present the average online performance of the separate actors of each ensemble agent – i.e. for each ensemble agent it is the average curve of the data collection performance of each actor in its ensemble. The figure below shows that each actor successfully trains in separation and the performance only breaks when the actors get combined into their ensembles.
>
> [ensembles-train.png](https://postimg.cc/bSbgngRc)
>
> This experiment supports the novelty of the ED2 algorithm as it is not enough to simply combine independent runs of SAC or SOP – neither by averaging policies nor by using voting. Our technical innovation is in showing how to successfully construct a state-of-the-art performing and stable algorithm using a straightforward setup and a relatively small number of RL tricks. In doing so, we tested a variety of setups, providing insight into what works and what does not, which is hard to come by. We argue that this work is valuable both for practitioners and researchers, and allows them to build on top of our work while avoiding pitfalls that we described and tested empirically.
>
> We address the minor problems in the next comment.

---

> > ### Author Response · Authors · 2021-11-22
> > **Response to the minor problems**
> >
> > Minor problem #1
> >
> > We point the reviewer to Appendix E Implementation Details where we discuss the action normalization and the ERE replay buffer details.
> >
> > Minor problem #2
> >
> > We understand that the method section might currently lack theoretical argumentation. Our method aims to exploit neural network biases stemming from its random initialization. During the data collection, one network is used throughout the whole episode and thus its bias induces temporary coherent behaviour, suitable for exploration. This is explained in Section 3 and will be elaborated more in the camera-ready paper.
> >
> > Minor problem #3
> >
> > We tried to classify each design choice and group them together. We put all the normalization techniques under one category, but we acknowledge that the action normalization could be considered as an exploration technique too.
> >
> > Minor problem #4
> >
> > The standard reward function as defined in Mujoco consists of several components. Some of them are designed to improve learning. However, they may obscure the true objective of moving forward. Reporting velocity is arguably more aligned with this goal. For instance, the return for Humanoid simply standing is close to 5000 (\approx 5 living reward * 1000 time steps). That being said, we understand such presentations can be thought of as “cherry-picked” as we don’t present similar metrics for other envs at the moment. Thus we have decided to remove it from the revised text to save space for other experiments.

---

### Official Review · Reviewer_thDH · 2021-11-08

**Correctness:** 4
**Technical Novelty And Significance:** 2
**Empirical Novelty And Significance:** 2
**Recommendation:** 5
**Confidence:** 3

**Main Review:**

The paper is very well-written, flows smoothly and is a pleasure to read. The ideas are well articulated and clear.

Result in Fig 2 is surprising as it suggests that the additive normal action noise is entirely unnecessary. This has been a fixture is most DRL algorithms. However, looking in Appendix B.2, the authors did not test Ornstein-Uhlenbeck noise (DDPG paper) as one of the baselines. Adding this popular choice would complete this empirical evaluation. I do not think that OU noise would change the conclusions, but it would be nice to include for the sake of completeness - considering that it is used in some of the seminal works in the field.

The results for the stability experiments in Fig 6-8 on inference stability, asymptotic performance stability and training stability do not seem that surprising to me. Wouldn’t an ensemble method fundamentally lead to more stable learning? SUNRISE seems to be an exception here for some domains. However, ED2 being more ‘stable’ that a lone network algorithm (like SAC baseline) run seems to be rather obvious to me. An ensemble method should fundamentally be more stable as it has the advantage of N=5 random initializations. Would perhaps a better comparison be against an ensemble of SAC runs?

The main contribution of the paper are (1) the empirical study into the various design choices within DRL algorithms used for continuous control settings and (2) an ensemble approach that integrates these learnings. While the ensemble method achieves SOTA results on many tasks and the empirical study presents some riveting results, the novelty in the paper is quite limited.


**Summary Of The Paper:**

The paper presents an empirical study evaluating the commonly accepted design choice in off-policy Deep RL algorithms in continuous control settings. The use of additive exploration noise, initialization choices, update frequency, and precision for retraining are tested empirically highlighting some interesting results. The paper also introduces ED2 - an ensemble method utilizing the design choices from the study which is demonstrated to achieve SOTA results on Mujoco benchmarks.

**Summary Of The Review:**

The paper presents some interesting learnings via a diligent evaluation of the design choices used in Deep RL algorithms for continuous control settings. The paper presents an ensemble method using these learnings and posts some SOTA results. I believe the community would benefit from these learnings and the resulting ED2 approach. However, I do note that the novelty for the method is quite limited as ED2 is fundamentally an ensemble of previous method (SOP) with well-evaluated design parameters for the target task of Mujoco benchmarks.

---

> ### Author Response · Authors · 2021-11-22
> **We kindly thank the reviewer for the feedback**
>
> We kindly thank the reviewer for the feedback.
>
> As suggested by the reviewer, we add SOP with Ornstein-Uhlenbeck noise (SOP OU noise) as one of the baselines to complete our study. The SOP OU noise: does better than SOP with normal noise (SOP) and on par with ED2 in Hopper; does better than SOP, but still worse than ED2 in Walker; does much worse than SOP, det. DOP, and ED2 on Ant; and does on par with SOP, and hence does worse than ED2, in Humanoid.
>
> [sop-ou-noise.png](https://postimg.cc/3kMXp2yY)
>
> For a fair comparison, we run additional experiments where we compare ED2 to the ensemble of SAC and SOP runs. Because of the short space of time, we present performance results for only some variants and with a limited number of seeds (at least 3 and up to 10 for each curve). However, they still present the effects we want to discuss. We will run the complete study comparing ED2 performance & stability to the ensemble SAC and SOP variants for the camera-ready paper.
>
> In the figures below, we present the ensemble of SAC runs performance (ESAC) and the ensemble of SOP runs performance (ESOP), where the actors are combined either by averaging the policies (mean policy) or by using voting described in the paper (vote policy), and compare them ED2 and non-ensemble SAC and SOP – conclusions are:
>
> * The voting policy in general does better than the mean policy. We hypothesise that because ESAC and ESOP train their agents without the shared replay buffer, the agents converge to different behaviour and averaging the policies make these different behaviours compete with each other. In the vote policy case, the ensemble of critics decides on only one best action from the five proposed by actors and this effect is not visible so much. We plan to test this hypothesis for the camera-ready paper.
> * Only ESAC mean policy and ESOP vote policy achieve the near ED2 performance on Hopper. Any other method, in any other environment, does worse or much worse than ED2.
> * Moreover, the ensemble versions of SAC and SOP often do worse even than the non-ensemble agents.
>
> [ensembles-test.png](https://postimg.cc/YGT8jZYz)
>
> [ensemble-vs-sac.png](https://postimg.cc/qtb1Hz7Y)
>
> [ensemble-vs-sop.png](https://postimg.cc/sQMwjsQ2)
>
> To support our results, we present the average online performance of the separate actors of each ensemble agent – i.e. for each ensemble agent it is the average curve of the data collection performance of each actor in its ensemble. The figure below shows that each actor successfully trains in separation and the performance only breaks when the actors get combined into their ensembles.
>
> [ensembles-train.png](https://postimg.cc/bSbgngRc)
>
> This experiment supports the novelty of the ED2 algorithm as it is not enough to simply combine independent runs of SAC or SOP – neither by averaging policies nor by using voting. Our technical innovation is in showing how to successfully construct a state-of-the-art performing and stable algorithm using a straightforward setup and a relatively small number of RL tricks. In doing so, we tested a variety of setups, providing insight into what works and what does not, which is hard to come by. We argue that this work is valuable both for practitioners and researchers, and allows them to build on top of our work while avoiding pitfalls that we described and tested empirically.

---

### Decision · Program_Chairs · 2022-01-20

**Decision:**

Reject

**Comment:**

This paper studies improving continuous control. The paper suggests a practical, beneficial combination approach that does well in the presented experiments. It also provides some overview and comparison over several recent insights in RL. While both are valuable, multiple reviewers had concerns that the paper has some limitations on both. In particular, the proposed ensemble approach is quite simple though valuable, and that reviewers generally felt that raised their expectations as to the strength of the empirical results which was not yet there. The reviewers’ provided a lot of detailed feedback that may be useful in revising the contribution.